# PARTICLE GUIDANCE: NON-I.I.D. DIVERSE SAMPLING WITH DIFFUSION MODELS

**Gabriele Corso**[*1]**, Yilun Xu**[1]**, Valentin de Bortoli**[2]**, Regina Barzilay**[1]**, Tommi Jaakkola**[1]
[1]CSAIL, Massachusetts Institute of Technology, [2]ENS, PSL University

## ABSTRACT

In light of the widespread success of generative models, a significant amount of research has gone into speeding up their sampling time. However, generative models are often sampled multiple times to obtain a diverse set incurring a cost that is orthogonal to sampling time. We tackle the question of how to improve diversity and sample efficiency by moving beyond the common assumption of independent samples. We propose *particle guidance*, an extension of diffusion-based generative sampling where a joint-particle time-evolving potential enforces diversity. We analyze theoretically the joint distribution that particle guidance generates, how to learn a potential that achieves optimal diversity, and the connections with methods in other disciplines. Empirically, we test the framework both in the setting of conditional image generation, where we are able to increase diversity without affecting quality, and molecular conformer generation, where we reduce the state-of-the-art median error by 13% on average.

## 1 INTRODUCTION

Deep generative modeling has become pervasive in many computational tasks across computer vision, natural language processing, physical sciences, and beyond. In many applications, these models are used to take a number of representative samples of some distribution of interest like Van Gogh's style paintings or the 3D conformers of a small molecule. Although independent samples drawn from a distribution will perfectly represent it in the limit of infinite samples, for a finite number, this may not be the optimal strategy. Therefore, while deep learning methods have so far largely focused on the task of taking independent identically distributed (I.I.D.) samples from some distribution, this paper examines how one can use deep generative models to take a finite number of samples that can better represent the distribution of interest.

In other fields where *finite-samples* approximations are critical, researchers have developed various techniques to tackle this challenge. In molecular simulations, several enhanced sampling methods, like metadynamics and replica exchange, have been proposed to sample diverse sets of low-energy structures and estimate free energies. In statistics, Stein Variational Gradient Descent (SVGD) is an iterative technique to match a distribution with a finite set of particles. However, these methods are not able to efficiently sample complex distributions like images.

Towards the goal of better *finite-samples* generative models, that combine the power of recent advances with sample efficiency, we propose a general framework for sampling sets of particles using diffusion models. This framework, which we call *particle guidance* (PG), is based on the use of a time-evolving potential to guide the inference process. We present two different strategies to instantiate this new framework: the first, *fixed potential particle guidance*, provides ready-to-use potentials that require no further training and have little inference overhead; the second, *learned potential particle guidance*, requires a training process but offers better control and theoretical guarantees.

The theoretical analysis of the framework leads us to two key results. On one hand, we obtain an expression for the joint marginal distribution of the sampled process when using any arbitrary guidance potential. On the other, we derive a simple objective one can use to train a model to learn a time-evolving potential that exactly samples from a joint distribution of interest. We show this provides optimal joint distribution given some diversity constraint and it can be adapted to the addition of further constraints such as the preservation of marginal distributions. Further, we also

---

[*]Correspondence to `gcorso@mit.edu`

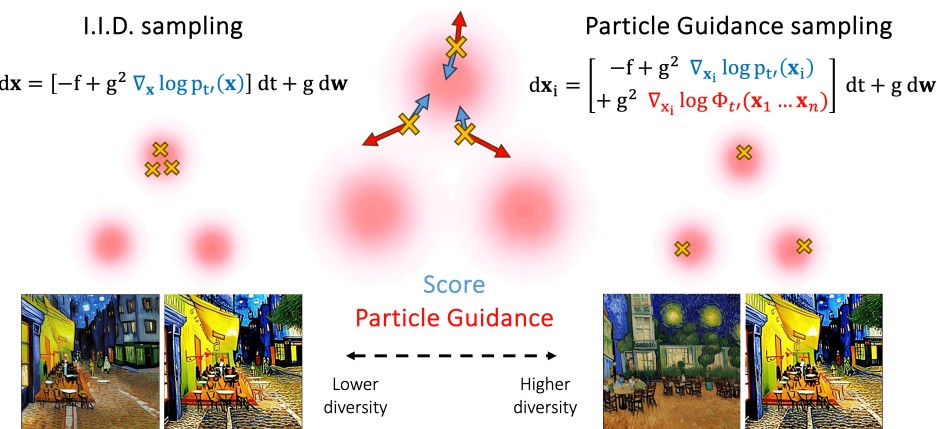

Figure 1: Comparison of I.I.D. and particle guidance sampling. The center figure represents each step, with the distribution in pink and the samples as yellow crosses, where particle guidance uses not only the score (in blue) but also the guidance from joint-potential (red), leading it to discover different modes (right-hand samples vs those on the left). At the bottom, Van Gogh cafe images samples generated with Stable Diffusion with and without particle guidance. A more detailed discussion on the suboptimality of I.I.D. sampling is presented in Appendix B.1.

demonstrate the relations of particle guidance to techniques for non-I.I.D. sampling developed in other fields and natural processes and discuss its advantages.

Empirically, we demonstrate the effectiveness of the method in both synthetic experiments and two of the most successful applications of diffusion models: text-to-image generation and molecular conformer generation. In the former, we show that particle guidance can improve the diversity of the samples generated with Stable Diffusion [Rombach et al., 2021] while maintaining a quality comparable to that of I.I.D. sampling. For molecular conformer generation, applied to the state-of-the-art method Torsional Diffusion [Jing et al., 2022], particle guidance is able to simultaneously improve precision and coverage, reducing their median error by respectively 19% and 8%. In all settings, we also study the critical effect that different potentials can have on the diversity and sample quality.

## 2 BACKGROUND

**Diffusion models** Let $p(x)$ be the data distribution we are interested in learning. Diffusion models [Song et al., 2021] define a forward diffusion process that has $p$ as the initial distribution and is described by $d\mathbf{x} = \mathbf{f}(\mathbf{x}, t)dt + g(t)d\mathbf{w}$, where $\mathbf{w}$ is the Wiener process. This forward diffusion process is then reversed using the corresponding reverse diffusion SDE $d\mathbf{x} = [-\mathbf{f}(\mathbf{x}, T - t) + g(T-t)^2 \nabla_\mathbf{x} \log p_{T-t}(\mathbf{x})]dt + g(T-t)d\mathbf{w}$ (using the forward time convention) where the evolving score $\nabla_\mathbf{x} \log p_t(\mathbf{x})$ is approximated with a learned function $s_\theta(\mathbf{x}, t)$.

One key advantage of diffusion models over the broad class of energy-based models [Teh et al., 2003] is their *finite-time sampling* property for taking a single sample. Intuitively, by using a set of smoothed-out probability distributions diffusion models are able to overcome energy barriers and sample every mode in finite time as guaranteed by the existence of the reverse diffusion SDE [Anderson, 1982]. In general, for the same order of discretization error, reverse diffusion SDE can efficiently sample from data distribution in much fewer steps than Langevin dynamics in energy-based models. For instance, Theorem 1 of Chen et al. [2022] shows that, assuming accurate learning of score, the convergence of diffusion SDE is independent of the isoperimetry constant of the target distribution. Langevin dynamics mixing speed can be exponentially slow if the spectral gap/isoperimetry constant is small. This critical property is orthogonal to the efficiency in the number of samples one needs to generate to cover a distribution; in this work, we aim to achieve sample efficiency while preserving the *finite-time sampling* of diffusion models.

Diffusion models were extended to Riemannian manifolds by De Bortoli et al. [2022], this formulation has found particular success [Jing et al., 2022; Corso et al., 2022; Yim et al., 2023] in scientific domains where data distributions often lie close to predefined submanifolds [Corso, 2023].

Classifier guidance (CG) [Dhariwal & Nichol, 2021] has been another technical development that has enabled the success of diffusion models on conditional image generation. Here a classifier $p_\theta(y|\mathbf{x}_t, t)$, trained to predict the probability of $\mathbf{x}_t$ being obtained from a sample of class $y$, is used to guide a conditional generation of class $y$ following:

$$d\mathbf{x} = [-\mathbf{f}(\mathbf{x}, t') + g(t')^2(s_\theta(\mathbf{x}, t') + \alpha \nabla_\mathbf{x} \log p_\theta(y|\mathbf{x}, t'))]dt + g(t')d\mathbf{w} \quad \text{where } t' = T - t$$

where $\alpha$ in theory should be 1, but, due to overspreading of the distribution, researchers often set it to larger values. This, however, often causes the collapse of the generation to a single or few modes, hurting the samples' diversity.

## 3 PARTICLE GUIDANCE

Our goal is to define a sampling process that promotes the diversity of a finite number of samples while retaining the advantages and flexibility that characterize diffusion models. Let $p(\mathbf{x})$ be some probability distribution of interest and $\nabla_\mathbf{x} \log p_t(\mathbf{x})$ be the score that we have learned to reverse the diffusion process $d\mathbf{x} = f(\mathbf{x}, t)dt + g(t)d\mathbf{w}$. Similarly to how classifier guidance is applied, we modify the reverse diffusion process by adding the gradient of a potential. However, we are now sampling together a whole set of particles $\mathbf{x}_1, ..., \mathbf{x}_n$, and the potential $\log \Phi_t$ is not only a function of the current point but a permutation invariant function of the whole set:

$$d\mathbf{x}_i = \left[ -f(\mathbf{x}_i, t') + g^2(t')\left( \nabla_{\mathbf{x}_i} \log p_{t'}(\mathbf{x}_i) + \nabla_{\mathbf{x}_i} \log \Phi_{t'}(\mathbf{x}_1, ..., \mathbf{x}_n) \right) \right]dt + g(t')d\mathbf{w}. \quad (1)$$

where the points are initially sampled I.I.D. from a prior distribution $p_T$. We call this idea *particle guidance* (PG). This framework allows one to impose different properties, such as diversity, on the set of particles being sampled without the need to retrain a new score model operating directly on the space of sets.

We will present and study two different instantiations of this framework:

1. **Fixed Potential PG** where the time-evolving joint potential is handcrafted, leading to very efficient sampling of diverse sets without the need for any additional training. We present this instantiation in Section 5 and show its effectiveness on critical real-world applications of diffusion models in Section 6.

2. **Learned Potential PG** where we learn the time-evolving joint potential to provably optimal joint distributions. Further, this enables direct control of important properties such as the preservation of marginal distributions. We present this instantiation in Section 7.

## 4 CONNECTIONS WITH EXISTING METHODS

As discussed in the introduction, other fields have developed methods to improve the tradeoff between sampling cost and coverage of the distribution of interest. In this section, we will briefly introduce four methods (coupled replicas, metadynamics, SVGD and electrostatics) and draw connections with *particle guidance*.

### 4.1 COUPLED REPLICAS AND METADYNAMICS

In many domains linked to biochemistry and material science, researchers study the properties of the physical systems by collecting several samples from their Boltzmann distributions using molecular dynamics or other enhanced sampling methods. Motivated by the significant cost that sampling each individual structure requires, researchers have developed a range of techniques to improve sample efficiency and speed by, for example, reducing the correlation of subsequent samples from slow-mixing Markov chains. The most popular of these techniques are parallel sampling with coupled replicas and sequential sampling with metadynamics.

As the name suggests, replica methods involve directly taking $n$ samples of a system with the different sampling processes, replicas, occurring in parallel. In particular, coupled replica methods

[Hummer & Köfinger, 2015; Pasarkar et al., 2023] create a dependency between the replicas by adding, like *particle guidance*, an extra potential $\Phi$ to the energy function to enforce diversity or match experimental observables. This results in energy-based sampling procedures that target:

$$\tilde{p}(\mathbf{x}_1, \ldots, \mathbf{x}_n) = \Phi(\mathbf{x}_1, \ldots, \mathbf{x}_n) \prod_{i=1}^{n} p(\mathbf{x}_i).$$

Metadynamics [Laio & Parrinello, 2002; Barducci et al., 2008] was also developed to more efficiently sample the Boltzmann distribution of a given system. Unlike replica methods and our approach, metadynamics is a sequential sampling technique where new samples are taken based on previously taken ones to ensure diversity, typically across certain collective variables of interest $s(\mathbf{x})$. In its original formulation, the Hamiltonian at the $k^{\text{th}}$ sample is augmented with a potential as:

$$\tilde{H}_k = H - \omega \sum_{j<k} \exp\left(-\frac{\|s(\mathbf{x}) - s(\mathbf{x}_j^0)\|^2}{2\sigma^2}\right)$$

where $H$ is the original Hamiltonian, $\mathbf{x}_j^0$ are the previously sampled elements and $\omega$ and $\sigma$ parameters set a priori. Once we take the gradient and perform Langevin dynamics to sample, we obtain dynamics that, with the exception of the fixed Hamiltonian, resemble those of *particle guidance* in Eq. 4 where

$$\nabla_{\mathbf{x}_i} \log \Phi_t(\mathbf{x}_1, \cdots, \mathbf{x}_n) \leftarrow \nabla_{\mathbf{x}_i} \omega \sum_{j<i} \exp\left(-\frac{\|s(\mathbf{x}_i) - s(\mathbf{x}_j^0)\|^2}{2\sigma^2}\right).$$

Although they differ in their parallel or sequential approach, both coupled replicas and metadynamics can be broadly classified as energy-based generative models. As seen here, energy-based models offer a simple way of controlling the joint distribution one converges to by simply adding a potential to the energy function. On the other hand, however, the methods typically employ an MCMC sampling procedure, which lacks the critical *finite-time sampling* property of diffusion models and significantly struggles to cover complex probability distributions such as those of larger molecules and biomolecular complexes. Additionally, the MCMC typically necessitates a substantial number of steps, generally proportional to a polynomial of the data dimension [Chewi et al., 2020]. With particle guidance, we instead aim to achieve both properties (controllable diversity and finite time sampling) at the same time. We can simulate the associated SDE/ODE with a total number of steps that is independent of the data dimension.

### 4.2 SVGD

Stein Variational Gradient Descent (SVGD) [Liu & Wang, 2016] is a well-established method in the variational inference community to iteratively transport a set of particles to match a target distribution. Given a set of initial particles $\{\mathbf{x}_1^0 \ldots \mathbf{x}_n^0\}$, it updates them at every iteration as:

$$\mathbf{x}_i^{\ell-1} \leftarrow \mathbf{x}_i^\ell + \epsilon_\ell \psi(\mathbf{x}_i^\ell) \quad \text{where} \quad \psi(\mathbf{x}) = \frac{1}{n-1} \sum_{j=1}^{n} [k(\mathbf{x}_j^\ell, \mathbf{x}) \nabla_{\mathbf{x}_j^\ell} \log p(\mathbf{x}_j^\ell) + \nabla_{\mathbf{x}_j^\ell} k(\mathbf{x}_j^\ell, \mathbf{x})] \quad (2)$$

where $k$ is some (similarity) kernel and $\epsilon_\ell$ the step size. Although SVGD was developed with the intent of sampling a set of particles that approximate some distribution $p$ without the direct goal of obtaining diverse samples, SVGD and our method have a close relation.

This relation between our method and SVGD can be best illustrated under specific choices for drift and potential under which the probability flow ODE discretization of *particle guidance* can be approximated as (derivation in Appendix A.5):

$$\mathbf{x}_i^{t+\Delta t} \approx \mathbf{x}_i^t + \epsilon_t(\mathbf{x}_i) \psi_t(\mathbf{x}_i^t) \quad \text{where} \quad \psi(\mathbf{x}) = \frac{1}{n-1} \sum_{j=1}^{n} [k_t(\mathbf{x}_j^t, \mathbf{x}) \nabla_{\mathbf{x}} \log p_t(\mathbf{x}) + \nabla_{\mathbf{x}_j^t} k_t(\mathbf{x}_j^t, \mathbf{x})] \quad (3)$$

Comparing this with Eq. 2, we can see a clear relation in the form of the two methods, with some key distinctions. Apart from the different constants, the two methods use different terms for the total score component. Interestingly both methods use smoothed-out scores, however, on the one hand, particle guidance uses the *diffused* score at the specific particle $\mathbf{x}_i$, $\nabla_{\mathbf{x}_i} \log p_t(\mathbf{x}_i)$, while on the other, SVGD smooths it out by taking a weighted average of the score of nearby particles weighted by the similarity kernel $(\sum_j k(\mathbf{x}_i, \mathbf{x}_j) \nabla_{\mathbf{x}_j} \log p(\mathbf{x}_j))/(\sum_j k(\mathbf{x}_i, \mathbf{x}_j))$.

The reliance of SVGD on other particles for the "smoothing of the score", however, causes two related problems, firstly, it does not have the *finite-time sampling* guarantee that the time evolution of diffusion models provides and, secondly, it suffers from the collapse to few local modes near the initialization and cannot discover isolated modes in data distribution [Wenliang & Kanagawa, 2020]. This challenge has been theoretically [Zhuo et al., 2018] and empirically [Zhang et al., 2020] studied with several works proposing practical solutions. In particular, relevant works use an annealing schedule to enhance exploration [D'Angelo & Fortuin, 2021] or use score matching to obtain a noise-conditioned kernel for SVGD [Chang et al., 2020]. Additionally, we empirically observe that the score smoothing in SVGD results in blurry samples in image generation.

### 4.3 ELECTROSTATICS

Recent works [Xu et al., 2022; 2023b] have shown promise in devising novel generative models inspired by the evolution of point charges in high-dimensional electric fields defined by the data distribution. It becomes natural therefore to ask whether particle guidance could be seen as describing the evolution of point charges when these are put in the same electric field such that they are not only attracted by the data distribution but also repel one another. One can show that this evolution can indeed be seen as the combination of Poisson Flow Generative Models with particle guidance, where the similarity kernel is the extension of Green's function in $N+1$-dimensional space, *i.e.*, $k(x, y) \propto 1/||x - y||^{N-1}$. We defer more details to Appendix A.6.

## 5 FIXED POTENTIAL PARTICLE GUIDANCE

In this section, we will present and study a simple, yet effective, instantiation of particle guidance based on the definition of the time-evolving potential as a combination of predefined kernels. As we will see in the experiments in Section 6, this leads to significant sample efficiency improvements with no additional training required and little inference overhead.

To promote diversity and sample efficiency, in our experiments, we choose the potential $\log \Phi_t$ to be the negative of the sum of a pairwise similarity kernel $k$ between each pair of particles $\log \Phi_t(\mathbf{x}_1, ... \mathbf{x}_n) = -\frac{\alpha_t}{2} \sum_{i,j} k_t(\mathbf{x}_i, \mathbf{x}_j)$ obtaining:

$$d\mathbf{x}_i = \left[ -f(\mathbf{x}_i, t') + g^2(t') \left( \nabla_{\mathbf{x}_i} \log p_{t'}(\mathbf{x}_i) - \alpha_{t'} \nabla_{\mathbf{x}_i} \sum_{j=1}^{n} k_{t'}(\mathbf{x}_i, \mathbf{x}_j) \right) \right] dt + g(t')d\mathbf{w} \quad (4)$$

Intuitively, the kernel term will push our different samples to be dissimilar from one another while at the same time the score term will try to match our distribution. Critically, this does not come at a significant additional runtime as, in most domains, the cost of running the pairwise similarity kernels is very small compared to the execution of the large score network architecture. Moreover, it allows the use of domain-specific similarity kernels and does not require training any additional classifier or score model. We can also view the particle guidance Eq. (4) as a sum of reverse-time SDE and a guidance term. Thus, to attain a more expedited generation speed, the reverse-time SDE can also be substituted with the probability flow ODE [Song et al., 2021].

### 5.1 THEORETICAL ANALYSIS

To understand the effect that *particle guidance* has beyond simple intuition, we study the joint distribution of sets of particles generated by the proposed reverse diffusion. However, unlike methods related to energy-based models (see coupled replicas, metadynamics, SVGD in Sec. 4) analyzing the effect of the addition of a time-evolving potential $\log \Phi_t$ in the reverse diffusion is non-trivial.

While the score component in *particle guidance* is the score of the sequence of probability distributions $\tilde{p}_t(\mathbf{x}_1, \ldots, \mathbf{x}_n) = \Phi_t(\mathbf{x}_1, \ldots, \mathbf{x}_n) \prod_{i=1}^{n} p_t(\mathbf{x}_i)$, we are not necessarily sampling exactly $\tilde{p}_0$ because, for an arbitrary time-evolving potential $\Phi_t$, this sequence of marginals does not correspond to a diffusion process. One strategy used by other works in similar situations [Du et al., 2023] relies on taking, after every step or at the end, a number of Langevin steps to reequilibrate and move the distribution back towards $\tilde{p}_t$. This, however, increases significantly the runtime cost (every Langevin step requires score evaluation) and is technically correct only in the limit of infinite steps leaving uncertainty in the real likelihood of our samples. Instead, in Theorem 1, we use the Feynman-Kac theorem to derive a formula for the exact reweighting that particle guidance has on a distribution (derivation in Appendix A.1):

**Theorem 1.** *Under integrability assumptions, sampling $\mathbf{x}_1^T, ..., \mathbf{x}_n^T$ from $p_T$ and following the particle guidance reverse diffusion process, we obtain samples from the following joint probability distribution at time $t = 0$:*

$$\hat{p}_0(\mathbf{x}_1, \ldots, \mathbf{x}_n) = \mathbb{E}[Z \exp[-\int_0^T g(t)^2 \{\langle \nabla \log \Phi_t(\mathbf{X}_t), \nabla \log \hat{p}_t(\mathbf{X}_t) \rangle + \Delta \log \Phi_t(\mathbf{X}_t)\} \mathrm{d}t]],$$

*with $Z$ (explicit in the appendix) such that*

$$\prod_{i=1}^N p_0(\mathbf{x}_i) = \mathbb{E}[Z],$$

*$(\mathbf{X}_t)_{t \in [0,T]}$ is a stochastic process driven by the equation*

$$\mathrm{d}\mathbf{X}_t = \{f(\mathbf{X}_t, t) - g(t)^2 \nabla \log p_t(\mathbf{X}_t)\} \mathrm{d}t + g(t)\mathrm{d}\mathbf{w}, \qquad \mathbf{X}_0 = \{\mathbf{x}_i\}_{i=1}^N.$$

Hence the density $\hat{p}_0$ can be understood as a reweighting of the random variable $Z$ that represents I.I.D. sampling.

**Riemannian Manifolds.**   Note that our theoretical insights can also be extended to the manifold framework. This is a direct consequence of the fact that the Feynman-Kac theorem can be extended to the manifold setting, see for instance Benton et al. [2022].

## 5.2   PRESERVING INVARIANCES

The objects that we learn to sample from with generative models often present invariances such as the permutation of the atoms in a molecule or the roto-translation of a conformer. To simplify the learning process and ensure these are respected, it is common practice to build such invariances in the model architecture. In the case of diffusion models, to obtain a distribution that is invariant to the action of some group $G$ such as that of rotations or permutations, it suffices to have an invariant prior and build a score model that is $G$-equivariant [Köhler et al., 2020; Xu et al., 2021]. Similarly, in our case, we are interested in distributions that are invariant to the action of $G$ on any of the set elements (see Section 6.2), we show that a sufficient condition for this invariance to be maintained is that the time-evolving potential $\Phi_t$ is itself invariant to $G$-transformations of any of its inputs (see Proposition 1 in Appendix A.4).

## 6   EXPERIMENTS

Fixed potential particle guidance can be implemented on top of any existing trained diffusion model with the only requirement of specifying the potential/kernel to be used in the domain. We present three sets of empirical results in three very diverse domains. First, in Appendix C, we work with a synthetic experiment formed by a two-dimensional Gaussian mixture model, where we can visually highlight some properties of the method. In this section instead, we consider text-to-image and molecular conformer generation, two important tasks where diffusion models have established new state-of-the-art performances, and show how, in each of these tasks, particle guidance can provide improvements in sample efficiency pushing the diversity-quality Pareto frontier. The code is available at `https://github.com/gcorso/particle-guidance`.

### 6.1   TEXT-TO-IMAGE GENERATION

In practice, the most prevalent text-to-image diffusion models, such as Stable Diffusion [Rombach et al., 2021] or Midjourney, generally constrain the output budget to four images per given prompt. Ideally, this set of four images should yield a diverse batch of samples for user selection. However, the currently predominant method of classifier-free guidance [Ho, 2022] tends to push the mini-batch samples towards a typical mode to enhance fidelity, at the expense of diversity.

To mitigate this, we apply the proposed particle guidance to text-to-image generation. We use Stable Diffusion v1.5, pre-trained on LAION-5B [Schuhmann et al., 2022] with a resolution of $512 \times 512$, as our testbed. We apply an Euler solver with 30 steps to solve for the ODE version of particle guidance. Following [Xu et al., 2023a], we use the validation set in COCO 2014 [Lin et al., 2014] for evaluation, and the CLIP [Hessel et al., 2021]/Aesthetic score [Team, 2022] (higher is better) to assess the text-image alignment/visual quality, respectively. To evaluate the diversity within each batch of generated images corresponding to a given prompt, we introduce the *in-batch similarity*

*score*. This metric represents the average pairwise cosine similarity of features within an image batch, utilizing the pre-trained DINO [Caron et al., 2021] as the feature extractor. Contrasting the FID score, the in-batch similarity score specifically measures the diversity of a batch of images generated for a given prompt. We use a classifier-free guidance scale from 6 to 10 to visualize the trade-off curve between the diversity and CLIP/Aesthetic score, in line with prior works [Xu et al., 2023a; Saharia et al., 2022]. For particle guidance, we implement the RBF kernel on the down-sampled pixel space (the latent space of the VAE encoder-) in Stable Diffusion, as well as the feature space of DINO. Please refer to Appendix E.1 for more experimental details.

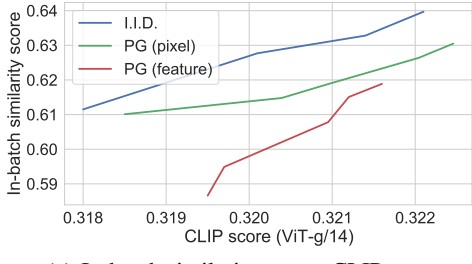
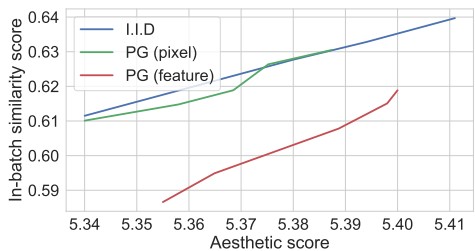

(a) In-batch similarity versus CLIP score

(b) In-batch similarity versus Aesthetic score

Figure 2: In-batch similiarity score versus **(a)** CLIP ViT-g/14 score and **(b)** Aesthetic score for text-to-image generation at $512 \times 512$ resolution, using Stable Diffusion v1.5 with a varying guidance scale from 6 to 10.

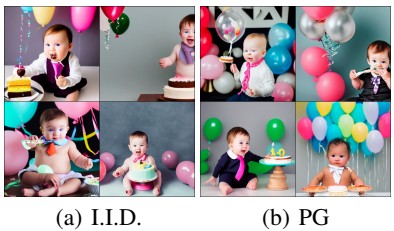
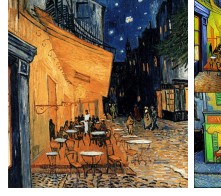
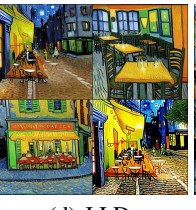
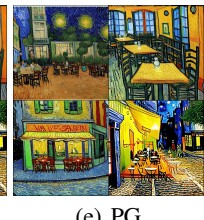

(a) I.I.D.          (b) PG          (c) Training data          (d) I.I.D.          (e) PG

Figure 3: Text prompt: (a,b) "A baby eating a cake with a tie around his neck with balloons in the background" (COCO); (c,d,e) "VAN GOGH CAFE TERASSE copy.jpg", with original training data in (c).

As shown in Fig. 2(a) and Fig. 2(b), particle guidance (PG) consistently obtains a better (lower) in-batch similarity score in most cases, given the same CLIP/Aesthetic score, with a classifier-free guidance scale ranging from 6 to 10. Conversely, we observe that while in-batch similarity score of I.I.D. sampling improves with the reduced classifier-free guidance scale, particle guidance continues to surpass I.I.D. sampling in terms of CLIP/Aesthetic score given the same in-batch similarity. When the potential is the similarity kernel applied in the feature space, particle guidance notably attains a lower in-batch similarity score compared to I.I.D. sampling or to the approach in the original downsampled pixel space. This suggests that utilizing a semantically meaningful feature space is more appropriate for determining distances between images.

In Fig. 3, we further visualize generated batches of four images per prompt by I.I.D. sampling and particle guidance (feature) with the same random seeds, when fixing the classifier-free guidance scale to 9. We can see that particle guidance improves the visual diversity in the generated batch. Interestingly, particle guidance can also help to alleviate the memorization issue of Stable Diffusion [Somepalli et al., 2023]. For example, given the text prompt of a painting from LAION dataset, particle guidance (Fig. 3(e)) avoids the multiple replications of the training data in the I.I.D. setting (the top-left and the bottom-right images in Fig. 3(d)). We provide extended samples in Appendix F, and additionally show that SVGD (Eq. 2) fails to promote diversity, instead yielding a set of blurry images.

## 6.2 MOLECULAR CONFORMER GENERATION

Molecular conformer generation is a key task in computational chemistry that consists of finding the set of different conformations that a molecule most likely takes in 3D space. Critically it is often important to find all or most of the low-energy conformers as each can determine a different behavior

Table 1: Quality of generated conformer ensembles for the GEOM-DRUGS test set in terms of Coverage (%) and Average Minimum RMSD (Å). We follow the experimental setup from [Ganea et al., 2021], for experimental details and introduction of the baselines please refer to Appendix D.

| | Recall | | | | Precision | | | |
|---|---|---|---|---|---|---|---|---|
| | Coverage ↑ | | AMR ↓ | | Coverage ↑ | | AMR ↓ | |
| Method | Mean | Med | Mean | Med | Mean | Med | Mean | Med |
| RDKit ETKDG | 38.4 | 28.6 | 1.058 | 1.002 | 40.9 | 30.8 | 0.995 | 0.895 |
| OMEGA | 53.4 | 54.6 | 0.841 | 0.762 | 40.5 | 33.3 | 0.946 | 0.854 |
| GeoMol | 44.6 | 41.4 | 0.875 | 0.834 | 43.0 | 36.4 | 0.928 | 0.841 |
| GeoDiff | 42.1 | 37.8 | 0.835 | 0.809 | 24.9 | 14.5 | 1.136 | 1.090 |
| Torsional Diffusion | 72.7 | 80.0 | 0.582 | 0.565 | 55.2 | 56.9 | 0.778 | 0.729 |
| TD w/ particle guidance | **77.0** | **82.6** | **0.543** | **0.520** | **68.9** | **78.1** | **0.656** | **0.594** |

(e.g. by binding to a protein). This necessity is reflected in the metrics used by the community that look both at coverage (also called recall) and precision over the set predictions.

Over the past few years, molecular conformer generation has been extensively studied by the machine learning community, with well-established benchmarks [Axelrod & Gomez-Bombarelli, 2022] and several generative models designed specifically for this task [Ganea et al., 2021; Xu et al., 2021; Jing et al., 2022]. However, all these methods are based on training a generative model to generate single samples and then running this model several times (more than 200 on average in the standard GEOM-DRUGS dataset) to generate a large number of I.I.D. samples.

As discussed before, however, this strategy is suboptimal to generate representative sets of samples and cover the distribution. Therefore, we take the state-of-the-art conformer generation model, *torsional diffusion*, and, without retraining the model itself, we show that we can obtain significant improvements in both coverage and precision via particle guidance.

*Torsional diffusion* [Jing et al., 2022] defines the diffusion process over the manifold defined by changes in torsion angles from some initial conformer because of the relative rigidity of the remaining degrees of freedom. Given this observation, we also define the guidance kernel on this manifold as an RBF kernel over the dihedral angle differences.

Another important consideration when dealing with molecular conformers is given by the permutation symmetries that characterize several molecules: conformers that appear different might be very similar under permutations of the order of the atoms that do not change the bond structure. To maximize the sample efficiency and avoid generating similar conformers, we make the kernel invariant to these transformations. For this, we employ the simple strategy to take the minimum value of the original kernel under the different perturbations (formalized in Appendix D).

Table 1 shows that by applying particle guidance to SDE-based reverse process of torsional diffusion (see Appendix D for details) we are able to balance coverage and precision being able to obtain, without retraining the model, significantly improved results on both metrics with 8% and 19% simultaneous reductions respectively in recall and precision median AMR.

## 7 LEARNED POTENTIAL PARTICLE GUIDANCE

While the fixed potential particle guidance seen so far is very effective in improving the diversity of samples with little overhead, it is hard to argue about the optimality of the resulting joint distribution. This is because of the complexity of the expression obtained in Theorem 1 and its dependence on the data distribution itself. Furthermore, in some domains, particularly in scientific applications, researchers need to control the distribution that they are sampling. This is necessary, for example, to apply correct importance weights or compute free energy differences. While Theorem 1 allows us to theoretically analyze properties of the distribution, the joint and marginal distributions remain largely intractable.

In this section, we analyze how we can sample from desired joint probability distribution by learning a tailored time-evolving potential for particle guidance. Using the maximum entropy theorem [Csiszár, 1975], we can show that the distribution satisfying a bound on the expected value of a

(diversity) metric $\Phi_0$ while minimizing the KL divergence with the independent distribution is:

$$\hat{p}_0(\mathbf{x}_1, ..., \mathbf{x}_n) \propto \Phi_0(\mathbf{x}_1, ..., \mathbf{x}_n)^{\beta(\alpha)} \prod_{i=1}^{n} p(\mathbf{x}_i) \tag{5}$$

where $\beta$ is a function of $\alpha$, the value of the bound on $E_{\hat{p}}[\log \Phi_0]$.

## 7.1 TRAINING PROCEDURE

We now have to learn a time-evolving potential $\Phi_t$ that when used as part of the particle guidance framework generates $\hat{p}_0$ (we assume $\Phi_0$ is chosen such that $\beta(\alpha) = 1$). To achieve this, we mandate that the generation process of particle guidance in Eq. 1 adheres to the sequence of marginals $\hat{p}_t(\mathbf{x}_1^t, ..., \mathbf{x}_n^t) = \Phi_t(\mathbf{x}_1^t, ..., \mathbf{x}_n^t) \prod_{i=1}^{n} p_t(\mathbf{x}_i^0)$ and learn $\Phi_t^\theta$ to satisfy this evolution. Under mild assumptions, using Doob $h$-theory (derivation in Appendix A.2), we show that we can learn the $\Phi_t^\theta$ by the following objective:

$$\theta^* = \arg \min_\theta \mathbb{E}_{\mathbf{x}_1^0, ..., \mathbf{x}_n^0 \sim p_0} \mathbb{E}_{\mathbf{x}_i^t \sim p_{t|0}(\cdot|\mathbf{x}_i^0)}[\|\Phi_0(\mathbf{x}_1^0, ..., \mathbf{x}_n^0) - \Phi_t^\theta(\mathbf{x}_1^t, ..., \mathbf{x}_n^t)\|^2] \tag{6}$$

where $p_{t|0}$ is the Gaussian perturbation kernel in diffusion models. Importantly, here the initial $\mathbf{x}_i^0$ are sampled independently from the data distribution so this training scheme can be easily executed in parallel to learning the score of $p_t$.

## 7.2 PRESERVING MARGINAL DISTRIBUTIONS

While the technique discussed in the previous section is optimal in the maximum entropy perspective, it does not (for arbitrary $\Phi_0$) preserve the marginal distributions of individual particles, i.e. marginalizing $\mathbf{x}_i$ over $\hat{p}$ does not recover $p$. Although not critical in many settings and not respected, for a finite number of particles, neither by the related methods in Section 4 nor by the fixed potential PG, this is an important property in some applications.

Using again the maximum entropy theorem, we can show that the distribution satisfying a bound on the expected value of a (diversity) metric $\Phi_0'$ and preserving the marginal distribution while minimizing the KL divergence with the independent distribution can be written as:

$$\hat{p}_0(\mathbf{x}_1, ..., \mathbf{x}_n) \propto \Phi_0'(\mathbf{x}_1, ..., \mathbf{x}_n)^{\beta(\alpha)} \prod_{i=1}^{n} p(\mathbf{x}_i) \gamma_\theta(\mathbf{x}_i) \tag{7}$$

for some scalar function over individual particles $\gamma_\theta$. In Appendix A.3, we derive a new training scheme to learn the parameters of $\gamma_\theta$. This relies on setting the normalization constant to an arbitrary positive value and learning values of $\theta$ that respect the marginals. Once $\gamma_\theta$ is learned, its parameters can be frozen and the training procedure of Eq. 6 can be started.

## 8 CONCLUSION

In this paper, we have analyzed how one can improve the sample efficiency of generative models by moving beyond I.I.D. sampling and enforcing diversity, a critical challenge in many real applications that has been largely unexplored. Our proposed framework, particle guidance, steers the sampling process of diffusion models toward more diverse sets of samples via the definition of a time-evolving joint potential. We have studied the theoretical properties of the framework such as the joint distribution it converges to for an arbitrary potential and how to learn potential functions that sample some given joint distribution achieving optimality and, if needed, preserving marginal distributions. Finally, we evaluated its performance in two important applications of diffusion models text-to-image generation and molecular conformer generation, and showed how in both cases it is able to push the Pareto frontier of sample diversity vs quality.

We hope that particle guidance can become a valuable tool for practitioners to ensure diversity and fair representation in existing tools even beyond the general definition of diversity directly tackling known biases of generative models. Further, we hope that our methodological and theoretical contributions can spark interest in the research community for better joint-particle sampling methods.

## ACKNOWLEDGMENTS

We thank Xiang Cheng, Bowen Jing, Timur Garipov, Shangyuan Tong, Renato Berlinghieri, Saro Passaro, Simon Olsson, and Hannes Stärk for their invaluable help with discussions and review of the manuscript. We also thank the anonymous reviewers for their useful feedback and suggestions. Concurrently with our work, Alex Pondaven also developed a similar idea[1].

This work was supported by the NSF Expeditions grant (award 1918839: Collaborative Research: Understanding the World Through Code), the Machine Learning for Pharmaceutical Discovery and Synthesis (MLPDS) consortium, the Abdul Latif Jameel Clinic for Machine Learning in Health, the DTRA Discovery of Medical Countermeasures Against New and Emerging (DOMANE) threats program, the DARPA Accelerated Molecular Discovery program, the NSF AI Institute CCF-2112665, the NSF Award 2134795, the GIST-MIT Research Collaboration grant, MIT-DSTA Singapore collaboration, and MIT-IBM Watson AI Lab.

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

# A  DERIVATIONS

## A.1  JOINT DISTRIBUTION UNDER PARTICLE GUIDANCE

In this section, we provide the proof of Theorem 1. First, we restate the Feynman-Kac theorem. Let $u : [0, T] \times \mathbb{R}^d$ such that for any $t \in [0, T]$ and $x \in \mathbb{R}^d$ we have

$$\partial_t u(t, x) + \langle b(t, x), \nabla u(t, x) \rangle + (1/2) \langle \Sigma(t, x), \nabla^2 u(t, x) \rangle - V(t, x) u(t, x) + f(t, x) = 0, \quad (8)$$

with $u(T, x) = \Phi(T, x)$. Then, under integrability and regularity assumptions, see Karatzas & Shreve [1991] for instance, we have

$$u(0, x) = \mathbb{E}[\int_0^T \exp[- \int_0^r V(\tau, \mathbf{X}_\tau) d\tau] f(r, \mathbf{X}_r) dr + \exp[- \int_0^T V(\tau, \mathbf{X}_\tau) d\tau] \Phi(T, \mathbf{X}_T) \mid \mathbf{X}_0 = x], \quad (9)$$

with $u(T, x) = \Phi(T, x)$ and $d\mathbf{X}_t = b(t, \mathbf{X}_t) dt + \Sigma(t, \mathbf{X}_t) d\mathbf{B}_t$. In the rest of this section, we derive the specific case of Theorem 1.

We recall that the generative model with particle guidance is given by $(\hat{p}_t)_{t \in [0, T]}$ and is associated with the generative model

$$d\hat{\mathbf{Y}}_t = \{-f(\hat{\mathbf{Y}}_t, T - t) + g(T - t)^2 (s_\theta(\hat{\mathbf{Y}}_t, T - t) + \nabla \log \Phi_{T-t}(\hat{\mathbf{Y}}_t))\} dt + g(T - t) d\mathbf{w}. \quad (10)$$

We also recall that the generative model without particle guidance is given by $(q_t)_{t \in [0, T]}$ and is associated with the generative model

$$d\mathbf{Y}_t = \{-f(\mathbf{Y}_t, T - t) + g(T - t)^2 s_\theta(\mathbf{Y}_t, T - t)\} dt + g(T - t) d\mathbf{w}. \quad (11)$$

Using the Fokker-Planck equation associated with equation 11 we have for any $x \in (\mathbb{R}^d)^N$

$$\partial_t q_t(x) + \text{div}(\{-f(T - t, \cdot) + g(T - t)^2 s_\theta(T - t, \cdot)\} q_t)(x) - (g(T - t)^2/2) \Delta q_t(x) = 0. \quad (12)$$

This can also be rewritten as

$$\partial_t q_t(x) + \langle -f(T - t, x) + g(T - t)^2 s_\theta(x, T - t), \nabla q_t(x) \rangle - (g(T - t)^2/2) \Delta q_t(x) \quad (13)$$

$$+ \text{div}(\{-f(\cdot, T - t) + g(T - t)^2 s_\theta(T - t, \cdot)\})(x) q_t(x) = 0 \quad (14)$$

Denoting $u_t = q_{T-t}$ we have

$$\partial_t u_t(x) + \langle f(x, t) - g(t)^2 s_\theta(x, t), \nabla u_t(x) \rangle + (g(t)^2/2) \Delta u_t(x) \quad (15)$$

$$- \text{div}(\{-f(t, \cdot) + g(t)^2 s_\theta(\cdot, t)\})(x) u_t(x) = 0. \quad (16)$$

Note that since $u_t = q_{T-t}$, we have that $u_t = p_t$ with the conventions from 3. Now combining this result with equation 8 and equation 9 with $V(t, x) = \text{div}(\{-f(\cdot, t) + g(t)^2 s_\theta(t, \cdot)\})(x)$ and $f = 0$ we have that

$$u_0(x) = \mathbb{E}[Z], \quad (17)$$

with

$$Z = \exp[- \int_0^T V(\tau, \mathbf{X}_\tau) d\tau] p_0(\mathbf{X}_T), \quad (18)$$

and

$$d\mathbf{X}_t = \{f(t, \mathbf{X}_t) - g(t)^2 s_\theta(\mathbf{X}_t, t)\} dt + g(t) d\mathbf{w}. \quad (19)$$

with $\mathbf{X}_0 = x$. We now consider a similar analysis in the case of the generative with particle guidance. Using the Fokker-Planck equation associated with equation 10 we have for any $x \in (\mathbb{R}^d)^N$

$$\partial_t \tilde{q}_t(x) + \text{div}(\{-f(\cdot, T-t) + g(T-t)^2 (s_\theta(\cdot, T-t) + \nabla \log \Phi_{T-t})\} \tilde{q}_t)(x) - (g(T-t)^2/2) \Delta \tilde{q}_t(x) = 0. \quad (20)$$

This can also be rewritten as

$$\partial_t \tilde{q}_t(x) + \langle -f(x, T - t) + g(T - t)^2 s_\theta(x, T - t), \nabla \tilde{q}_t(x) \rangle - (g(T - t)^2/2) \Delta \tilde{q}_t(x) \quad (21)$$

$$+ \text{div}(\{-f(\cdot, T - t) + g(T - t)^2 s_\theta(\cdot, T - t)\})(x) \tilde{q}_t(x) \quad (22)$$

$$+ g(T - t)^2 (\langle \log \Phi_{T-t}(x), \nabla \log \tilde{q}_t(x) \rangle + \Delta \log \Phi_{T-t}(x)) \tilde{q}_t(x) = 0. \quad (23)$$

Denoting $\hat{u}_t = \tilde{q}_{T-t}$ we have

$$\partial_t \hat{u}_t(x) + \langle f(t, x) - g(t)^2 s_\theta(x, t), \nabla \hat{u}_t(x) \rangle + (g(t)^2/2) \Delta \hat{u}_t(x) \quad (24)$$

$$- \text{div}(\{-f(\cdot, t) + g(t)^2 s_\theta(t, \cdot)\})(x) \hat{u}_t(x) \quad (25)$$

$$- g(t)^2 (\langle \nabla \log \Phi_t(x), \nabla \log \tilde{q}_{T-t}(x) \rangle + \Delta \log \Phi_t(x)) \hat{u}_t(x) = 0. \quad (26)$$

Following the convention of 3, we have that $\hat{u}_t = \hat{p}_0$. Now combining this result with equation 8 and equation 9 with $\hat{V}(t, x) = \text{div}(\{-f(\cdot, t) + g(t)^2 s_\theta(\cdot, t)\})(x) + g(t)^2(\langle \nabla \log \Phi_t(x), \nabla \log \tilde{p}_{T-t}(x) \rangle + \Delta \log \Phi_t(x))$ and $f = 0$ we have that

$$\hat{u}_0(x) = \mathbb{E}[\hat{Z}], \tag{27}$$

with

$$\hat{Z} = \exp[-\int_0^T \hat{V}(\tau, \mathbf{X}_\tau)\mathrm{d}\tau]p_0(\mathbf{X}_T), \tag{28}$$

and

$$\mathrm{d}\mathbf{X}_t = \{f(\mathbf{X}_t, t) - g(t)^2 s_\theta(\mathbf{X}_t, t)\}\mathrm{d}t + g(t)\mathrm{d}\mathbf{w}. \tag{29}$$

again with $\mathbf{X}_0 = x$. We conclude the proof upon noting that

$$\hat{Z} = Z \exp[-\int_0^T g(t)^2(\langle \nabla \log \Phi_t(\mathbf{X}_t), \nabla \log \hat{u}_t(\mathbf{X}_t) \rangle + \Delta \log \Phi_t(\mathbf{X}_t))dt]. \tag{30}$$

**Interpretation** An interpretation of this reweighting term can be given through the lens of SVGD. We introduce the *Stein operator* as in Liu & Wang [2016] given by for any $\Psi : (\mathbb{R}^d)^N \to (\mathbb{R}^d)^N$ by

$$\mathcal{A}_{\hat{p}_t}(\Psi_t) = \nabla \log \hat{p}_t \Psi_t^\top + \nabla \Psi_t. \tag{31}$$

Using $\Psi_t = \nabla \log \Phi_t$, we get that

$$\text{Tr}(\mathcal{A}_{\hat{p}_t}(\Psi_t)) = \langle \nabla \log \Phi_t, \nabla \log \hat{p}_t \rangle + \Delta \log \Phi_t. \tag{32}$$

The squared expectation of this quantity w.r.t. a distribution $q$ on $(\mathbb{R}^d)^N$ is the *Kernel Stein Discrepancy* (KSD) between $q$ and $\hat{p}_t$ given the kernel $\log \Phi_t$.

## A.2 SAMPLING A PREDEFINED JOINT DISTRIBUTION

For ease of derivation via the Doob h-transform, we temporarily reverse the time from $t$ to $T - t$. Here, $p_T$ is treated as the data distribution, and $\Phi_T$ is regarded as the potential, as specified by users. We now consider another model. Namely, we are looking for a generative model $\hat{p}_t$ with $t \in [0, T]$ such that for any $t \in [0, T]$ we have $\hat{p}_t = p_t \Phi_t$ with $\Phi_T$ given by the user. In layman's terms, this means that we are considering a *factorized* model for all times $t$ with the additional requirement that at the final time $T$, the model is given by $p_T = \hat{p}_T \Phi_T$ with $\Phi_T$ known. This is to be compared with Theorem 1. Indeed in Theorem 1 while the update on the generative dynamics is explicit (particle guidance term), the update on the density is not. In what follows, we are going to see, using tools from Doob $h$-transform theory, that we can obtain an expression for the update of the drift in the generative process when considering models of the form $\hat{p}_t = p_t \Phi_t$.

More precisely, we consider the following model. Let $\hat{p}_T = p_T$ and for any $s, t \in [0, T]$ with $s < t$ and $\mathbf{x}_s^{1:n} = \{\mathbf{x}_s^i\}_{i=1}^n \in (\mathbb{R}^d)^n$ and $\mathbf{x}_t^{1:n} = \{\mathbf{x}_t^i\}_{i=1}^n \in (\mathbb{R}^d)^n$ we define

$$\hat{p}_{t|s}(\mathbf{x}_t^{1:n}|\mathbf{x}_s^{1:n}) = p_{t|s}(\mathbf{x}_t^{1:n}|\mathbf{x}_s^{1:n})\Phi_t(\mathbf{x}_t^{1:n})/\Phi_s(\mathbf{x}_s^{1:n}), \tag{33}$$

with $\Phi_t$ which satisfies for any $\mathbf{x}_t^{1:n} \in (\mathbb{R}^d)^n$

$$\partial_t \Phi_t(\mathbf{x}_t^{1:n}) + \langle -f_{T-t}(\mathbf{x}_t^{1:n}) + g(T-t)^2 \nabla \log p_t(\mathbf{x}_t^{1:n}), \nabla \Phi_t(\mathbf{x}_t^{1:n}) \rangle + (g(T-t)^2/2)\Delta \Phi_t(\mathbf{x}_t^{1:n}) = 0, \tag{34}$$

with $\Phi_T$ given. Note that equation 34 expresses that $\Phi_t$ satisfies the backward Kolmogorov equation. Under mild assumptions, using Doob $h$-theory, we get that there exists $(\hat{\mathbf{X}}_t)_{t \in [0,T]}$ such that for any $t \in [0, T]$ we have $\hat{\mathbf{Y}}_t \sim \hat{p}_t$ and for any $t \in [0, T]$

$$\mathrm{d}\hat{\mathbf{Y}}_t = \{-f_{T-t}(\hat{\mathbf{Y}}_t) + g(T-t)^2[\nabla \log p_t(\hat{\mathbf{Y}}_t) + \nabla \log \Phi_t(\hat{\mathbf{Y}}_t)]\}\mathrm{d}t + g(T-t)\mathrm{d}\mathbf{w}. \tag{35}$$

The main difficulty is to compute $\Phi_t$ for any $t \in [0, T]$. Under mild assumptions, solutions to the backward Kolmogorov equation 34 are for any $t \in [0, T]$ by

$$\Phi_t(\mathbf{x}_t^{1:n}) = \mathbb{E}[\Phi_T(\mathbf{Y}_T)|\mathbf{Y}_t = \mathbf{x}_t^{1:n}] = \int \Phi_T(\mathbf{Y}_T = \mathbf{x}_T^{1:n})p_{T|t}(\mathbf{x}_T^{1:n}|\mathbf{x}_t^{1:n})d\mathbf{x}_T^{1:n}, \tag{36}$$

where we have

$$\mathrm{d}\mathbf{Y}_t = \{-f_{T-t}(\mathbf{Y}_t) + g(T-t)^2 \nabla \log p_t(\mathbf{Y}_t)\}\mathrm{d}t + g(T-t)\mathrm{d}\mathbf{w}. \tag{37}$$

This means that $(\mathbf{Y}_t)_{t \in [0,T]}$ is given by the original generative model, with time-dependent marginals $p_t$. The expression equation 36, suggests to parameterize $\Phi_t$ by $\Phi_t^\theta$ and to consider the loss function

$$\ell_t(\theta) = \mathbb{E}_{\mathbf{Y}_T} \mathbb{E}_{\mathbf{Y}_t \sim p_{t|T}(\cdot|\mathbf{Y}_T)}[\|\Phi_T(\mathbf{Y}_T) - \Phi_t^\theta(\mathbf{Y}_t)\|^2]. \tag{38}$$

Then, we can define a global loss function $\mathcal{L}(\theta) = \int_0^T \lambda(t)\ell_t(\theta)\mathrm{d}t$ where $\lambda_t$ is some weight. One problem with this original loss function is that it requires sampling and integrating with respect to $\mathbf{Y}_t$ which requires sampling from the generative model.

Recall that we reverse the time from $t$ to $T - t$ at the beginning. Reverse back to the original convention in the main text, Eq. (39) can be expressed as

$$\ell_t(\theta) = \mathbb{E}_{\mathbf{X}_0 \sim p_0} \mathbb{E}_{\mathbf{X}_t \sim p_{t|0}(\cdot|\mathbf{X}_T)}[\|\Phi_0(\mathbf{X}_0) - \Phi_t^\theta(\mathbf{X}_t)\|^2]. \tag{39}$$

### A.3 PRESERVING MARGINAL DISTRIBUTIONS

For arbitrary time evolving potentials $\Phi_t(\mathbf{x}_1, ..., \mathbf{x}_n)$ sampling using particle guidance does not preserve the marginals $p(\mathbf{x}_i) \neq \int_{\mathbf{x}_1,...,\mathbf{x}_{i-1},\mathbf{x}_{i+1},...,\mathbf{x}_n} \hat{p}(\mathbf{x}_1, ..., \mathbf{x}_n)d\mathbf{x}_1...d\mathbf{x}_{i-1}d\mathbf{x}_{i+1}...d\mathbf{x}_n$. In many domains this is not required and none of the methods discussed in Section 4 have this property for finite number of particles, however, in domains where, for example, one wants to obtain unbiased estimates of some function this property may be useful.

While the technique discussed in Section 7 allows us to use any potential $\Phi_0(\mathbf{x}_1, ..., \mathbf{x}_n)$ choosing $\Phi_0$ in such a way that preserves marginals is hard for non-trivial potentials and distributions. Therefore, we propose to learn a non-trivial marginal preserving $\Phi_0^\theta$ from the data in the following way. Let $\Phi_0^\theta(\mathbf{x}_1, ..., \mathbf{x}_n) = \Phi_0'(\mathbf{x}_1, ..., \mathbf{x}_n) \prod_i \gamma_\theta(\mathbf{x}_i)$ where $\Phi_0'$ is some predefined joint potential that, for example, encourages diversity in the joint distribution and $\gamma_\theta$ is a learned scalar function operating on individual points that counterbalances the effect that $\Phi_0'$ has on marginals while maintaining its effect on sample diversity.

How do we learn such scaling function $\gamma_\theta$ to preserve marginals? By definition, the individual marginal distribution is (consider $i = 1$ w.l.o.g.):

$$\begin{aligned}
\hat{p}_1(\mathbf{x}_1) &= \frac{\int_{\mathbf{x}_2,...,\mathbf{x}_n} \Phi_0'(\mathbf{x}_1, ..., \mathbf{x}_n) \prod_i p(\mathbf{x}_i)\gamma_\theta(\mathbf{x}_i)d\mathbf{x}_2...d\mathbf{x}_n}{\int_{\mathbf{x}_1',...,\mathbf{x}_n'} \Phi_0'(\mathbf{x}_1', ..., \mathbf{x}_n') \prod_i p(\mathbf{x}_i')\gamma_\theta(\mathbf{x}_i')d\mathbf{x}_1'...d\mathbf{x}_n'} = \\
&= \frac{p(\mathbf{x}_1)E_{\mathbf{x}_2,...,\mathbf{x}_n \sim \prod_{i>1} p(\mathbf{x}_i)}\big[\Phi_0'(\mathbf{x}_1, ..., \mathbf{x}_n) \prod_i \gamma_\theta(\mathbf{x}_i)\big]}{E_{\mathbf{x}_1',...,\mathbf{x}_n' \sim \prod_i p(\mathbf{x}_i')}\big[\Phi_0'(\mathbf{x}_1', ..., \mathbf{x}_n') \prod_i \gamma_\theta(\mathbf{x}_i')\big]} = \\
&= p(\mathbf{x}_1) \frac{E_{\mathbf{x}_2,...,\mathbf{x}_n \sim \prod_{i>1} p(\mathbf{x}_i)}\big[\Phi_0^\theta(\mathbf{x}_1, ..., \mathbf{x}_n)\big]}{E_{\mathbf{x}_1',...,\mathbf{x}_n' \sim \prod_i p(\mathbf{x}_i')}\big[\Phi_0^\theta(\mathbf{x}_1', ..., \mathbf{x}_n')\big]}
\end{aligned}$$

Therefore $\hat{p}_1 = p$ if and only if the fraction is always equal to 1 (intuitively for the marginal to be maintained on average the potential should have no effect). Assuming that the joint potential $\Phi_0'$ is invariant to the permutation to its inputs. Then one can also prove that, $\hat{p}_1 = p$ if and only if $E_{\mathbf{x}_2,...,\mathbf{x}_n \sim \prod_{i>1} p(\mathbf{x}_i)}\big[\Phi_0^\theta(\mathbf{x}_1, ..., \mathbf{x}_n)\big]$ is equal to a positive constant $C$ for any $x_1$. We can then minimize the following regression loss to learn the scalar function $\gamma_\theta$, by matching $E_{\mathbf{x}_2,...,\mathbf{x}_n \sim \prod_{i>1} p(\mathbf{x}_i)}\big[\Phi_0^\theta(\mathbf{x}_1, ..., \mathbf{x}_n)\big]$ and $C$:

$$\min_\theta \quad E_{\mathbf{x}_1}(E_{\mathbf{x}_2,...,\mathbf{x}_n \sim \prod_i p(\mathbf{x}_i)}\big[\Phi_0'(\mathbf{x}_1, ..., \mathbf{x}_n) \prod_{i \neq 1} \gamma_\theta(\mathbf{x}_i)\big] - \frac{C}{\gamma_\theta(\mathbf{x}_1)})^2$$

However, achieving this objective necessitates costly Monte Carlo estimations for the expectation over $n - 1$ independent samples. Moreover, obtaining an unbiased estimator for the full-batch gradient in a mini-batch setup is challenging. To bypass this issue, we implement a greedy update rule that optimizes the value $\gamma_\theta(\mathbf{x}_1)$ on the single sample $\mathbf{x}_1$. The greedy update can be viewed as continuous extension of the Iterative Proportional Fitting (IPF) for symmetry matrices. The essence of the IPF algorithm lies in determining scaling factors for each row and column of a given matrix, ensuring that the sum of each row and column (the marginals) aligns with a specified target value. IPF is known for its uniqueness and convergence guarantees [Sinkhorn, 1964]. Let's denote

$\phi_i(\mathbf{x}_1, ..., \mathbf{x}_n) = \texttt{stop\_grad}(\Phi_0'(\mathbf{x}_1, ..., \mathbf{x}_n) \prod_{j \neq i} \gamma_\theta(\mathbf{x}_j))$, the objective for the greedy update rule is as follows:

$$\min_{\gamma_\theta(\mathbf{x}_1)} \quad E_{\mathbf{x}_1}(E_{\mathbf{x}_2,...,\mathbf{x}_n \sim \prod_{i \neq 1} p(\mathbf{x}_i)} \phi_1(\mathbf{x}_1, ..., \mathbf{x}_n) - \frac{C}{\gamma_\theta(\mathbf{x}_1)})^2$$

The gradient w.r.t $\theta$ in the objective above is:

$$\theta' = \theta - \beta E_{\mathbf{x}_1, \mathbf{x}_2, ..., \mathbf{x}_n \sim \prod_i p(\mathbf{x}_i)} \left[ \frac{2C}{\gamma_\theta^2(\mathbf{x}_1)} \left( \phi_1(\mathbf{x}_1, ..., \mathbf{x}_n) - \frac{C}{\gamma_\theta(\mathbf{x}_1)} \right) \right] \nabla_\theta \gamma_\theta(x_1) \quad (40)$$

where $\beta$ is the learning rate. The corresponding update by stochastic gradient is:

$$\theta' = \theta - \beta \frac{1}{n} \sum_{i=1}^{n} \left[ \frac{2C}{\gamma_\theta^2(\mathbf{x}_i)} \left( \phi_i(\mathbf{x}_1, ..., \mathbf{x}_n) - \frac{C}{\gamma_\theta(\mathbf{x}_i)} \right) \right] \nabla_\theta \gamma_\theta(x_i) \quad (41)$$

$$= \theta - \beta \frac{1}{n} \sum_{i=1}^{n} \left[ \frac{2C}{\gamma_\theta^3(\mathbf{x}_i)} \left( \Phi_0^\theta(\mathbf{x}_1, ..., \mathbf{x}_n) - C \right) \right] \nabla_\theta \gamma_\theta(x_i) \quad (42)$$

for $\mathbf{x}_1, ..., \mathbf{x}_n \sim \prod_i p(\mathbf{x}_i)$. The stochastic gradient is an unbiased estimator of the gradient in Eq. (40).

### A.3.1 EMPIRICAL SYNTHETIC EXPERIMENTS

We demonstrate this training paradigm in a synthetic experiment using a mixture of Gaussian distributions in 2D. In particular, we set $p$ to be a mixture of 7 Gaussians with the same variance with placed as shown in Figure 4.A. The middle Gaussian has a weight that is four times that of the others.

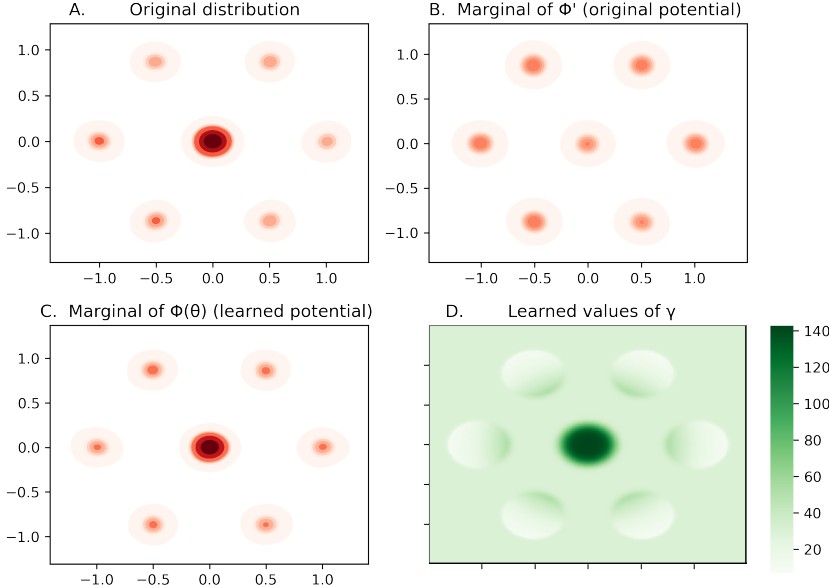

Figure 4: Synthetic experiment on learning a potential that preserves marginal distributions. The description of each plot can be found in the text.

Figure 4.B shows the marginal distribution when sampling 10 particles with joint distribution:

$$\tilde{p}_0(\mathbf{x}_1, ..., \mathbf{x}_n) \propto \Phi_0'(\mathbf{x}_1, ..., \mathbf{x}_n) \prod_{i=1}^{n} p(\mathbf{x}_i)$$

where $\Phi_0'$ is a measure of diversity that in this case we take to be the exponential of the negative of the mean of pairwise Euclidean RBF similarity kernels. Table 2 highlights how sampling from

this modified joint distribution significantly increases the diversity of the samples, especially when it comes to the expected value of $\log \Phi'_0$ itself. However, as clear from Figure 4.B, the marginal distribution is significantly changed with the middle mode being sampled only 13% of the times instead of 40% and even the shape of the outer modes being altered.

To fix this we learn a function $\gamma_\theta$, here parameterized simply as the set of values on a grid representing the domain. We follow the training scheme presented in Eq. (41) obtaining the function presented in Figure 4.D. As evident from the plot, this has the effect of overweighing samples in the central mode, while downsampling samples in the outside modes, especially when coming from the outer parts. In Figure 4.C we plot the marginal of the resulting distribution:

$$\hat{p}_0(\mathbf{x}_1, ..., \mathbf{x}_n) \propto \Phi'_0(\mathbf{x}_1, ..., \mathbf{x}_n) \prod_{i=1}^n p(\mathbf{x}_i) \gamma_\theta(\mathbf{x}_i).$$

Although the marginals closely match, $\hat{p}_0$ still has more diversity in its sets of samples compared to I.I.D. sampling of $p$, however, as seen in Table 2 the level of diversity in $\hat{p}_0$ is lower than that of $\tilde{p}_0$, highlighting the "cost" of imposing the preservation of marginals.

Table 2: Values of different observables under different joint probability distributions. For every method we take 5000 samples (of 10 particles), samples from $\tilde{p}_0$ and $\hat{p}_0$ were obtained reweighting 50000 samples of the independent I.I.D. distribution.

| Observable | I.I.D. $p$ | $\tilde{p}_0$ | $\hat{p}_0$ |
|---|---|---|---|
| Number of modes recovered at every sample | 4.9 | 5.9 | 5.3 |
| Expected value of $\log \Phi'_0$ | -37.3 | -31.8 | -36.3 |

### A.4 INVARIANCE OF PARTICLE GUIDANCE

**Proposition 1.** *Let $G$ be the group of rotations or permutations of a set of vectors. Assuming that $p_T(\mathbf{x})$ is a $G$-invariant distribution, the learned score $s(\mathbf{x}, t)$ and $f(\mathbf{x}, t)$ are $G$-equivariant and the potential $\log \Phi_t(\mathbf{x}_1 \ldots \mathbf{x}_n)$ is $G$-invariant to a transformation of any of its inputs, then the resulting distribution we sample from will also be $G$-invariant to a transformation of any of the elements of the set.*

Note that in this section we will derive this specific formulation for the group of rotations or permutations and the Brownian motion in Euclidean space. For a more general statement on Lie groups $G$ and Brownian motions associated with a given metric, one could generalize the result from Yim et al. [2023] Proposition F.2.

*Proof.* For simplicity, we will consider Euler discretization steps going with time from $T$ to 0 (as used in our experiments), however, the proposition applies in the continuous setting too:

$$p_\theta(\mathbf{x}_i^{(t-1)}|\mathbf{x}_{1:n}^{(t)}) = p_{\mathbf{z}}(\mathbf{x}_i^{(t-1)} - \mathbf{x}_i^{(t)} + f(\mathbf{x}_i^{(t)}, t) - g^2(\mathbf{s}_\theta(\mathbf{x}_i^{(t)}, t) + \nabla_{\mathbf{x}_i^{(t)}} \log \Phi_t(\mathbf{x}_{1:n}^{(t)})))$$

where $\mathbf{z} \sim N(0, g^2 I)$. Without loss of generality since the whole method is invariant to permutations of the particles, consider $\mathbf{x}_n$ to be the particle to which we apply $T_g$ the transformation of an arbitrary group element $g$.

Since by assumption $\log \Phi_t(\mathbf{x}_{1:n}^{(t)}) = \log \Phi_t(\mathbf{x}_{1:n-1}^{(t)}, T_g(\mathbf{x}_n^{(t)}))$ we have $p_\theta(\mathbf{x}_i^{(t-1)}|\mathbf{x}_{1:n}^{(t)}) = p_\theta(\mathbf{x}_i^{(t-1)}|\mathbf{x}_{1:n-1}^{(t)}, T_g(\mathbf{x}_n^{(t)}))$.

On the other hand, since $\log \Phi_t(\mathbf{x}_{1:n}^{(t)})$ is invariant to $G$ transformations of $\mathbf{x}_n^{(t)}$, its gradient w.r.t. the same variable will be $G$-equivariant. Therefore:

$$p_\theta(T_g(\mathbf{x}_n^{(t-1)})|\mathbf{x}_{1:n-1}^{(t)}, T_g(\mathbf{x}_n^{(t)})) =$$
$$= p_{\mathbf{z}}(T_g(\mathbf{x}_n^{(t-1)}) - T_g(\mathbf{x}_n^{(t)}) + f(T_g(\mathbf{x}_n^{(t)}), t) - g^2(\mathbf{s}_\theta(T_g(\mathbf{x}_n^{(t)}), t) + \nabla_{\mathbf{x}_n^{(t)}} \log \Phi_t(\mathbf{x}_{1:n-1}^{(t)}, T_g(\mathbf{x}_n^{(t)}))))$$
$$= p_{\mathbf{z}}(T_g(\mathbf{x}_n^{(t-1)}) - T_g(\mathbf{x}_n^{(t)}) + T_g(f(\mathbf{x}_n^{(t)}, t)) - g^2(T_g(\mathbf{s}_\theta(\mathbf{x}_n^{(t)}, t)) + T_g(\nabla_{\mathbf{x}_n^{(t)}} \log \Phi_t(\mathbf{x}_{1:n}^{(t)}))))$$
$$= p_{\mathbf{z}}(T_g(\mathbf{x}_n^{(t-1)} - \mathbf{x}_n^{(t)} + f(\mathbf{x}_n^{(t)}, t) - g^2(\mathbf{s}_\theta(\mathbf{x}_n^{(t)}, t) + \nabla_{\mathbf{x}_n^{(t)}} \log \Phi_t(\mathbf{x}_{1:n}^{(t)})))) = p_\theta(\mathbf{x}_n^{(t-1)}|\mathbf{x}_{1:n}^{(t)})$$

where between lines 2 and 3 we have used the equivariance assumptions and in the latter two the properties of elements of $G$.

Putting these together, we follow a similar derivation of Proposition 1 from Xu et al. [2021]:

$$
\begin{aligned}
&p_\theta(\mathbf{x}_{1:n-1}^{(0)}, T_g(\mathbf{x}_n^{(0)})) = \\
&= \int p(\mathbf{x}_{1:n-1}^{(T)}, T_g(\mathbf{x}_n^{(T)})) \prod_{t=1}^{T} p_\theta(\mathbf{x}_{1:n-1}^{(t-1)}, T_g(\mathbf{x}_n^{(t-1)})|\mathbf{x}_{1:n-1}^{(t)}, T_g(\mathbf{x}_n^{(t)})) = \\
&= \int \left( \prod_{i<n} p(\mathbf{x}_i^{(T)}) \right) \left( \prod_{t=1}^{T} \prod_{i<n} p_\theta(\mathbf{x}_i^{(t-1)}|\mathbf{x}_{1:n-1}^{(t)}, T_g(\mathbf{x}_n^{(t)})) \right) \cdot \\
&\quad \cdot \left( p(T_g(\mathbf{x}_n^{(T)})) \prod_{t=1}^{T} p_\theta(T_g(\mathbf{x}_n^{(t-1)})|\mathbf{x}_{1:n-1}^{(t)}, T_g(\mathbf{x}_n^{(t)})) \right) = \\
&= \int \left( \prod_{i<n} p(\mathbf{x}_i^{(T)}) \right) \left( \prod_{t=1}^{T} \prod_{i<n} p_\theta(\mathbf{x}_i^{(t-1)}|\mathbf{x}_{1:n}^{(t)}) \right) \left( p(\mathbf{x}_n^{(T)}) \prod_{t=1}^{T} p_\theta(\mathbf{x}_n^{(t-1)}|\mathbf{x}_{1:n}^{(t)}) \right) = p_\theta(\mathbf{x}_{1:n}^{(0)})
\end{aligned}
$$

$\square$

## A.5 Particle Guidance as SVGD

In this section, we derive the approximation of Eq. 3 starting from the probability flow ODE equivalent of Eq. 4 under the assumptions of no drift $f(x,t) = 0$ and using the following form for $\Phi_t(x_1, ..., x_n) = (\sum_{i,j} k_t(x_i, x_j))^{-\frac{n-1}{2}}$ where $k_t$ is a similarity kernel based on the Euclidean distance (e.g. RBF kernel).

$$
dx_i = \left[ f(x_i, t) - \frac{1}{2} g^2(t) \left( \nabla_{x_i} \log p_t(x_i) + \nabla_{x_i} \log \left( \sum_{ij} k_t(x_i, x_j) \right)^{-\frac{n-1}{2}} \right) \right] dt
$$

$$
= -\frac{1}{2} g^2(t) dt \left( \nabla_{x_i} \log p_t(x_i) - \frac{\frac{1}{2} \nabla_{x_i} \sum_{ij} k_t(x_i, x_j)}{\frac{1}{n-1} \sum_{ij} k_t(x_i, x_j)} \right)
$$

Now we can simplify the numerator using the fact that $k_t$ is symmetric and approximate the denominator assuming that different particles will have similar average distances to other particles:

$$
\approx -\frac{1}{2} g^2(t) dt \left( \nabla_{x_i} \log p_t(x_i) - \frac{\nabla_{x_i} \sum_j k_t(x_i, x_j)}{\sum_j k_t(x_i, x_j)} \right)
$$

$$
= -\frac{g^2(t) dt}{2 S(x_i)} \left( \sum_j k_t(x_i, x_j) \nabla_{x_i} \log p_t(x_i) - \nabla_{x_i} k_t(x_i, x_j) \right)
$$

where $S(x_i) = \sum_j k_t(x_i, x_j)$. Now we can use the fact that $\nabla_{x_i} k_t(x_i, x_j) = -\nabla_{x_j} k_t(x_i, x_j)$ because the kernel only depends on the Euclidean distance between the two points:

$$
= -\frac{n \, g^2(t) dt}{2 S(x_i)} \left( \frac{1}{n-1} \sum_j k_t(x_i, x_j) \nabla_{x_i} \log p_t(x_i) + \nabla_{x_j} k_t(x_i, x_j) \right)
$$

Letting $\epsilon_t(x_i) = \frac{n \, g^2(t) \Delta t}{2 S(x_i)}$, we obtain Eq. 3:

$$
x_i^{t-\Delta t} \approx x_i^t + \epsilon_t(x_i) \psi_t(x_i^t) \quad \text{where} \quad \psi(x) = \frac{1}{n-1} \sum_{j=1}^{n} [k_t(x_j^t, x) \nabla_x \log p_t(x) + \nabla_{x_j^t} k_t(x_j^t, x)]
$$

### A.6 PARTICLE GUIDANCE IN POISSON FLOW GENERATIVE MODELS

In this section, we consider the more general Poisson Flow Generative Models++ [Xu et al., 2023b] framework in which the $N$-dimensional data distribution is embedded into $N + D$-dimensional space, where $D$ is a positive integer ($D = 1/D \to \infty$ recover PFGM [Xu et al., 2022]/diffusion models). The data distribution is interpreted as a positive charge distribution. Each particle independently follows the electric field generated by the $N$-dimensional data distribution $p(x)$ embedded in a $N + D$-dimensional space. One can similarly do particle guidance in the PFGM++ scenarios, treating the group of particles as negative charges, not only attracted by the data distribution but also exerting the mutually repulsive force. Formally, for the augmented data the ODE in PFGM++ (Eq.4 in Xu et al. [2023b]) is

$$\frac{dx}{dr} = \frac{E(x,r)_x}{E(x,r)_r}$$

where $E_x$ and $E_r$ are the electric fields for different coordinates:

$$E(x,r)_x = \frac{1}{S_{N+D-1}(1)} \int \frac{x-y}{(\|x-y\|^2 + r^2)^{\frac{N+D}{2}}} p(y) dy$$

$$E(x,r)_r = \frac{1}{S_{N+D-1}(1)} \int \frac{r}{(\|x-y\|^2 + r^2)^{\frac{N+D}{2}}} p(y) dy$$

Note that when $r = \sigma\sqrt{D}, D \to \infty$, the ODE is $\frac{dx}{dr} = \frac{E(x,r)_x}{E(x,r)_r} = -\frac{\sigma}{\sqrt{D}} \nabla_x \log p_\sigma(x)$ and the framework degenerates to diffusion models.

Now if we consider the repulsive forces among a set of (uniformly weighted) particles with the same anchor variables $r$, $\{(x_i, r)\}_{i=1}^n$, only the electric field in the $x$ coordinate changes (the component in the $r$ coordinate is zero). Denote the new electric field in $x$ component as $\hat{E}_x$:

$$\hat{E}(x_i,r)_x = \underbrace{E(x_i,r)_x}_{\text{attractive force by data}} + \underbrace{\frac{1}{S_{N+D-1}(1)} \frac{1}{n-1} \sum_{j \neq i} \frac{x_j - x_i}{(\|x_j - x_i\|^2)^{\frac{N+D}{2}}}}_{\text{repulsive force between particles}}$$

The corresponding new ODE for the $i$-th particle is

$$\frac{dx_i}{dr} = \frac{\hat{E}(x_i,r)_x}{E(x_i,r)_r}$$

$$= \frac{E(x_i,r)_x}{E(x_i,r)_r} + \frac{\frac{1}{S_{N+D-1}(1)} \frac{1}{n-1} \sum_{j \neq i} \frac{x_j - x_i}{(\|x_j - x_i\|^2)^{\frac{N+D}{2}}}}{E(x_i,r)_r}$$

$$= \underbrace{\frac{E(x_i,r)_x}{E(x_i,r)_r}}_{\text{predicted by pre-trained models}} + \underbrace{\frac{\frac{1}{n-1} \sum_{j \neq i} \frac{x_j - x_i}{(\|x_j - x_i\|^2)^{\frac{N+D}{2}}}}{\int \frac{r}{(\|x-y\|^2 + r^2)^{\frac{N+D}{2}}} p(y) dy}}_{\text{particle guidance}}$$

$$= \frac{E(x_i,r)_x}{E(x_i,r)_r} + \frac{\frac{1}{n-1} \sum_{j \neq i} \frac{x_j - x_i}{\|x_j - x_i\|^{N+D}}}{\frac{S_{N+D-1}}{r^{D-1} S_{D-1}} p_r(x_i)}$$

where $p_r$ is the intermidate distribution, and $S_n$ is the surface area of $n$-sphere. Clearly, the direction of the guidance term can be regarded as the sum of the gradient of $N + D$-dimensional Green's function $G(x,y) \propto 1/\|x-y\|^{N+D-2}$, up to some scaling factors:

$$\nabla_{x_i} G(x_i, x_j) = \frac{x_i - x_j}{\|x_j - x_i\|^{N+D}}$$

## A.7 COMBINATORIAL ANALYSIS OF SYNTHETIC EXPERIMENTS

**Proposition 2.** *Let us have a random variable taking a value equiprobably between N distinct bins. The expectation of the proportion of bins discovered (i.e. sampled at least once) after N samples is $1 - (\frac{N-1}{N})^N$ which tends to $1 - 1/e$ as N tends to infinity.*

*Proof.* Let $n_i$ be the number of samples in the $i^{\text{it}}$ bin. The proportion of discovered bins is equal to:

$$\frac{1}{N}\mathbb{E}\left[\sum_{i=1}^{N} I_{n_i>0}\right] = \frac{1}{N}\sum_{i=1}^{N}\mathbb{E}[I_{n_i>0}] = P(n_i > 0) = 1 - P(n_i = 0) = 1 - \left(\frac{N-1}{N}\right)^N$$

In the limit of $N \to \infty$:

$$\lim_{N\to\infty} 1 - \left(\frac{N-1}{N}\right)^N = 1 - y = 1 - \frac{1}{e}$$

since (using L'Hôpital's rule):

$$\log y = \lim_{N\to\infty} N \log\left(\frac{N-1}{N}\right) = \lim_{N\to\infty} \frac{\log(\frac{N-1}{N})}{1/N} = \lim_{N\to\infty} \frac{1/N^2}{-1/N^2} = -1$$

$\square$

Therefore for $N = 10$ we would expect $10 * (1 - 0.9^{10}) \approx 6.51$, which corresponds to what is observed in the empirical results of Section C.

**Proposition 3.** *(Coupon collector's problem) Let us have a random variable taking a value equiprobably between N distinct bins. The expectation of the number of samples required to discover all the bins is $N H_N$, where $H_N$ is the $N^{th}$ harmonic number, which is $\Theta(N \log N)$ as N tends to infinity.*

*Proof.* Len $L_{i|j}$ be the number of samples it takes to go from $j$ to $i$ bins discovered. We are therefore interested in $\mathbb{E}[L_{N|0}]$.

$$\mathbb{E}[L_{j|j-1}] = \frac{N-(j-1)}{N} * 1 + \frac{j-1}{N}\left[\mathbb{E}[L_{j|j-1}] + 1\right] \implies \mathbb{E}[L_{j|j-1}] = \frac{N}{N-(j-1)}$$

Therefore:

$$\mathbb{E}[L_{N|0}] = \mathbb{E}\left[\sum_{j=1}^{N} L_{j|j-1}\right] = \sum_{j=1}^{N}\mathbb{E}[L_{j|j-1}] = N\sum_{j=1}^{N}\frac{1}{N-(j-1)} = N\sum_{j=1}^{N}\frac{1}{j} = N H_N$$

Since $H_N$ is $\Theta(\log N)$, then $\mathbb{E}[L_{N|0}]$ is $\Theta(N \log N)$. $\square$

For $N = 10$, $\mathbb{E}[L_{10|0}] \approx 29.29$.

## B DISCUSSIONS

### B.1 SUBOPTIMALITY OF I.I.D. SAMPLING

**Combinatorial analysis** Let us consider again the setting of a random variable taking a value equiprobably between N distinct bins. In Fig. 5, we plot the expected number of modes (or bins) captured as a function of the number of steps as derived in Appendix A.7. This suggests that the region where I.I.D. sampling is considerably suboptimal in these regards (capturing the modes of the distribution) is when the number of samples is comparable with the number of modes: if the number of modes is much larger than I.I.D. samples are still likely to capture separate modes, and if the number of samples is much larger the number of uncaptured modes is likely to be small.

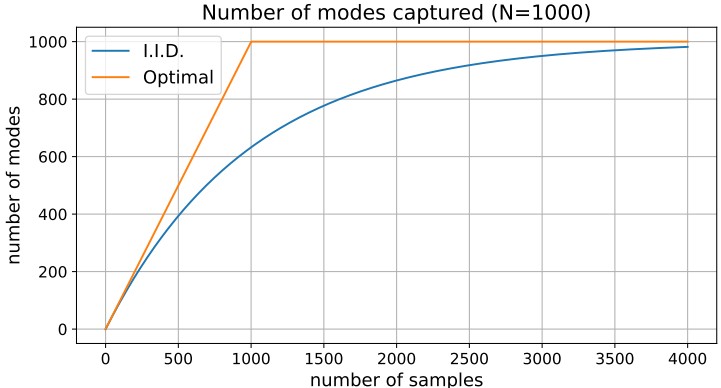

Figure 5: Plot of the functions $y = N(1 - (\frac{N-1}{N})^x)$ and $y = \min(x, N)$ for $N = 1000$ representing, respectively, the expected number of modes captured by I.I.D. sampling a distribution with N equiprobable modes and the optimal coverage.

**Literature** There is a vast literature that has studied the suboptimality of I.I.D. sampling from a statistical perspective and proposed different solutions. For example, in the field of Bayesian inference, antithetic sampling [Geweke, 1988] has been proposed as a way to reduce the variance of Monte Carlo estimates. Determinantal Point Processes [Kulesza et al., 2012] have also been widely studied as a technique to improve the diversity of samples.

## B.2 RUNTIME AND MEMORY OVERHEAD

The runtime overhead due to the addition of particle guidance to the inference procedure largely depends on the potential that is used and on the size of the set $n$. In particular, while computing kernels tends to be significantly cheaper than running the score model, the number of kernel computations scales quadratically with $n$ while the number of score model executions at every step is $n$.

In terms of memory, particle guidance does not create any significant overhead since the kernel computations can be aggregated per element. However, when running inference on GPU if $n$ is larger than the batch size that fits the GPU memory when running score model inference, the data might have to be moved back and forth between RAM and GPU memory to enable synchronous steps for particle guidance, causing further overhead.

In the case of our experiments on Stable Diffusion, the number of samples extracted (4) does not create significant overhead. However, in the setting of conformer generation on DRUGS different molecules can have very different numbers of conformers with some even having thousands of them. For efficiency, we therefore cap the size of $n$ in particle guidance to 128 and perform batches of 128 samples until the total number of conformers is satisfied.

**Other limitations** A badly chosen potential or in general one with a guidance weight too high can overly change the marginal likelihood and negatively impact the sampling quality. As an example in Fig. 6 the use of a particle guidance parameter four times larger than the best one caused various aliasing artifacts on the image.

However, in fixed potential particle guidance, its parameters can be easily fit at inference time, therefore, it is typically relatively inexpensive to test the optimal value of the guidance for the application of interest and the chosen potential. This leads to the prevention of too high guidance weights and the simple detection of bad potential when the optimal weight is close to 0.

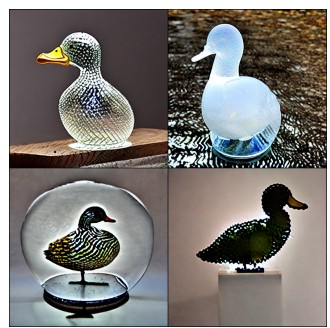

Figure 6: Example of a too large PG weight causing aliasing artifacts.

## C  Synthetic Experiments

To show visually the properties of particle guidance and its effect on sample efficiency, we use a two-dimensional Gaussian mixture model. In particular, we consider a mixture of $N = 10$ identical Gaussian distributions whose centers are equally spaced over the unit circle and whose variance is 0.005. These Gaussians form a set of approximately disjoint equal bins. As we are interested in inference, no model is trained and the true score of the distribution is given as an oracle.

As expected if one runs normal I.I.D. diffusion, the sample falls in one bin at random. Taking ten samples, as shown in Fig. 7, some of them will fall in the same bin and some bins will be left unfound. The empirical experiments confirm the combinatorial analysis (see Appendix A.7) which shows that the expected number of bins discovered with $N = 10$ samples is only 6.5 and it takes on average more than 29 samples to discover all the bins.

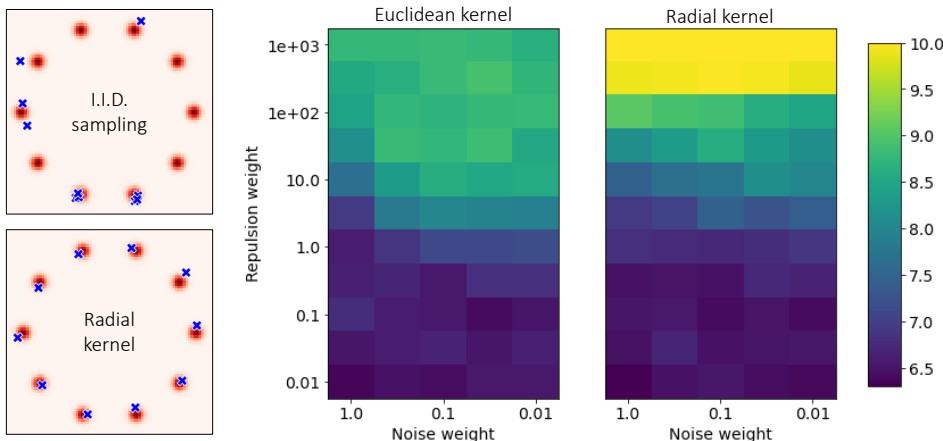

Figure 7: Left: plot of random samples (in blue) of the two-dimensional Gaussian mixture distribution (density depicted in red). I.I.D. samples often recover the same modes, while particle guidance with a radial kernel captures all modes. Right: average number of modes recovered with 10 samples as a function of the weight given by the diffusion noising terms and the potential weight when using an RBF kernel with Euclidean and radial distances respectively. As expected with little weight to the potential terms we obtain approximately 6.5 modes recovered in line with the I.I.D. diffusion performance. Further increasing the potential weight on the Euclidean creates instability.

In many settings this behavior is suboptimal, and we would want our model to discover all the modes of the distribution with as few samples as possible. Using the straightforward application of particle guidance with a simple RBF kernel based on the squared Euclidean distance, we are able to encourage diversity obtaining, on average, the discovery of nearly 9 bins on average from 10 samples (see Fig. 7).

Intrinsic diffusion models [Corso, 2023] have shown significant improvements when diffusion models operate on the submanifold where the data lies. Similarly, here building into the kernel the degrees of freedom over which the diversity lies helps the particle guidance to effectively distribute the samples over the distribution. We know that the different modes are distributed in a radial fashion, and thus we build an RBF kernel based on the angle difference w.r.t. the origin. Using this lower-dimensional kernel enables us to consistently discover all modes of the distribution. This submanifold observation aligns well with the practice of methods such as metadynamics where the kernels are defined over some lower-dimensional collective variables of interest.

## D  Molecular Conformer Generation Experiments

### D.1  Dataset, Metrics and Baselines

**Dataset**  We evaluate the method for the task of molecular conformer generation using the data from GEOM [Axelrod & Gomez-Bombarelli, 2022], a collection of datasets that has become the standard benchmark for this task in the machine learning community. In particular, we focus on

GEOM-DRUGS, the largest, most pharmaceutically relevant and widely used dataset in GEOM which consists of 304k drug-like molecules. For each of these molecules, an ensemble of conformers was generated with metadynamics in CREST [Pracht et al., 2020], a procedure that gives accurate structures but is prohibitive in high-throughput applications, costing an average of 90 core-hours per molecule. To be able to use existing pretrained models we rely on the experimental setup and splits introduced by Ganea et al. [2021] and used by several papers afterward. As we do not retrain the score model, we do not use the training set, instead, we finetune the inference parameters for *particle guidance* and the other ablation experiments on a random subset of 200 molecules out of 30433 from the validation set.

**Evaluation metrics** To evaluate conformer generation methods we want to test the ability of a method to generate a set of conformers that are both individually good poses (precision) and as a set cover the distribution of true conformers (recall). For this we employ the same evaluation setup and metrics used by several papers in the field starting from Ganea et al. [2021]. In this setup, methods are asked to generate twice as many conformers as in the original ensemble and then the so-called Average Minimum RMSD (AMR) and Coverage (COV) are measured for precision (P) and recall (R). For $K = 2L$ let $\{C_l^*\}_{l \in [1,L]}$ and $\{C_k\}_{k \in [1,K]}$ be respectively the sets of ground truth and generated conformers:

$$
\begin{aligned}
\text{COV-R} &:= \frac{1}{L} \left| \{l \in [1..L] : \exists k \in [1..K], \text{RMSD}(C_k, C_l^*) < \delta \} \right| \\
\text{AMR-R} &:= \frac{1}{L} \sum_{l \in [1..L]} \min_{k \in [1..K]} \text{RMSD}(C_k, C_l^*)
\end{aligned}
\tag{43}
$$

where $\delta$ is the coverage threshold (set to 0.75Å for the GEOM-DRUGS experiments). Swapping ground truth and generated conformers in the equations above we obtain the precision metrics.

**Baselines** As baselines we report the performances of previous methods as measured by Ganea et al. [2021] and Jing et al. [2022]. Cheminformatics conformer prediction methods rely on rules and heuristics derived from chemical structures to fix the local degrees of freedom and then use a combination of search and template techniques to set the more flexible degrees of freedom like torsion angles. The most accurate and widely used such methods include the open-source software RDKit ETKDG [Landrum et al., 2013] and the commercial tool OMEGA [Trott & Olson, 2010; Hawkins & Nicholls, 2012].

Before the already introduced Torsional Diffusion [Jing et al., 2022], a number of other machine learning approaches were proposed for this task, among these: GeoMol [Ganea et al., 2021] uses a GNN to sample directly from a random seed local structures around each atom and then torsion angles, GeoDiff [Xu et al., 2021] defines a equivariant diffusion model over atom coordinates, and CGCF [Shi et al., 2021] learns an energy-based model over the space of pairwise distance matrices.

### D.2 PARTICLE GUIDANCE SETUP

**Reverse diffusion** As discussed in Section 6.2, we applied particle guidance to torsional diffusion, as this is currently considered to be state-of-the-art and it uses, like most ML-based methods before, I.I.D. sampling during inference. We define the particle guidance kernel to operate directly on the implicit hypertorus manifold where torsional diffusion defines the diffusion process, this, at the same time, makes the kernel lower dimensional and it involves a minor modification to the existing inference procedure. The reverse diffusion process that we apply is:

$$
d\boldsymbol{\tau}_i = \underbrace{\frac{1}{2} g^2(T-t) \, \mathbf{s}(\boldsymbol{\tau}_i, L, T-t) \, dt}_{\text{diffusion ODE}} + \underbrace{\beta_{T-t}\left(\frac{1}{2} g^2(t) \, \mathbf{s}(\boldsymbol{\tau}_i, L, T-t) \, dt + g(T-t) \, d\mathbf{w}\right)}_{\text{Langevin diffusion SDE}}
$$
$$
+ \underbrace{\frac{\gamma_{T-t}}{2} g^2(T-t) \, \nabla_{\boldsymbol{\tau}_i} \log \Phi_{T-t}(\boldsymbol{\tau}_1, ..., \boldsymbol{\tau}_n) dt}_{\text{particle guidance}}
$$

where we follow the idea from Karras et al. [2022] of dividing the different components of the reverse diffusion and tuning their individual parameters. The potential was chosen to be:

$$\log \Phi_t(\boldsymbol{\tau}_1, ... \boldsymbol{\tau}_n) = -\frac{\alpha_t}{2n} \sum_{i,j} k_t(\boldsymbol{\tau}_i, \boldsymbol{\tau}_j) \quad \text{where} \quad k_t(\boldsymbol{\tau}_i, \boldsymbol{\tau}_j) = \exp(-\frac{||\boldsymbol{\tau}_i - \boldsymbol{\tau}_j||^2}{h_t}) \qquad (44)$$

where the difference of each torsion angle is computed to be in $(-\pi, \pi]$. $\alpha_t$, $\beta_t$, $\gamma_t$ and $h_t$ are inference hyperparameters that are 'logarithmically interpolated' between two end values chosen with hyperparameter tuning (using $T = 1$), e.g. $\alpha_t = \exp(t \log(\alpha_1) + (1 - t) \log(\alpha_0))$.

**Permutation invariant kernel**  Since the kernel operates on the torsion angle differences it is naturally invariant to SE(3) transformations, i.e. translations or rotations, of the conformers in space. Moreover, as illustrated in Fig. 2 of Jing et al. [2022], while exact torsion angle values depend on arbitrary choices of neighbors or orientation (to compute the dihedral angle) differences in torsion angles are invariant to these choices. However, one transformation that the kernel in Equation 44 is not invariant to are permutations of the atoms in the molecule. Many of these permutations lead to isomorphic molecular graphs where however each of the torsion angles may now refer to a different dihedral. To maximize the sample efficiency we make the kernel invariant to these by taking the minimum over the values of the kernel under all such permutations:

$$k_t'(\boldsymbol{\tau}_i, \boldsymbol{\tau}_j) = \min_{\pi \in \Pi} k_t(\boldsymbol{\tau}_i, P_\pi \boldsymbol{\tau}_j)$$

where $\Pi$ is the set of all permutations that keep the graph isomorphic (but do change the torsion angles assignment) and $P_\pi$ is the permutation matrix corresponding to some permutation $\pi$. In practice, these isomorphisms can be precomputed efficiently, however, to limit the overhead from applying the kernel multiple times, whenever there are more than 32 isomorphic graphs leading to a change in dihedral assignments we subsample these to only keep 32.

**Batch size**  The number of conformers one has to generate is given, for every molecule, by the benchmark (2L) and can vary significantly. To avoid significant computational overheads, we use batches of up to $n = 128$ until all the conformers for that particular molecule are generated.

### D.3  FULL RESULTS

We provide in Table 3 again the results reported in Table 1 with the additions of other baselines and ablation experiments. In particular, as ablations, on top of running non-invariant particle guidance i.e. without the minimization over the permutations described in the previous section, we also test low-temperature sampling, another variation of the inference-time procedure that has been proposed for diffusion models that we applied as described below.

**Low-temperature sampling.**  Low-temperature sampling of some distribution $p(\mathbf{x})$ with temperature $\lambda^{-1} < 1$ consists of sampling the distribution $p_\lambda(\mathbf{x}) \propto p(\mathbf{x})^\lambda$. This helps mitigate the overdispersion problem by concentrating more on high-likelihood modes and trading off sample diversity for quality. Exact low-temperature sampling is intractable for diffusion models, however, various approximation schemes exist. We use an adaptation of Hybrid Langevin-Reverse Time SDE proposed by Ingraham et al. [2022]:

$$d\boldsymbol{\tau} = -\left(\lambda_t + \frac{\lambda \psi}{2}\right) \mathbf{s}_{\theta, G}(C, t) \, g^2(t) \, dt + \sqrt{1 + \psi} \, g(t) \, d\mathbf{w} \quad \text{with} \quad \lambda_t = \frac{\sigma_d + \sigma_t}{\sigma_d + \sigma_t/\lambda}$$

where $\lambda$ (the inverse temperature), $\psi$ and $\sigma_d$ are parameters that can be tuned.

## E  EXPERIMENTAL DETAILS ON STABLE DIFFUSION

### E.1  SETUP

In this section, we detail the experimental setup on Stable Diffusion. We replace the score function ($\nabla_{\mathbf{x}_i} \log p_{t'}(\mathbf{x}_i)$) in the original particle guidance formula (Eq. (4)) with the classifier-free guidance formula [Ho, 2022]:

$$\tilde{s}(\mathbf{x}_i, c, t') = w \nabla_{\mathbf{x}_i} \log p_{t'}(\mathbf{x}_i, c) + (1 - w) \nabla_{\mathbf{x}_i} \log p_{t'}(\mathbf{x}_i)$$

Table 3: Quality of generated conformer ensembles for the GEOM-DRUGS test set in terms of Coverage (%) and Average Minimum RMSD (Å). Minimizing recall and precision refers to the hyperparameter choices that minimize the respective median AMR on the validation set.

| | Recall | | | | Precision | | | |
| | Coverage ↑ | | AMR ↓ | | Coverage ↑ | | AMR ↓ | |
| Method | Mean | Med | Mean | Med | Mean | Med | Mean | Med |
|---|---|---|---|---|---|---|---|---|
| RDKit ETKDG | 38.4 | 28.6 | 1.058 | 1.002 | 40.9 | 30.8 | 0.995 | 0.895 |
| OMEGA | 53.4 | 54.6 | 0.841 | 0.762 | 40.5 | 33.3 | 0.946 | 0.854 |
| CGCF | 7.6 | 0.0 | 1.247 | 1.225 | 3.4 | 0.0 | 1.837 | 1.830 |
| GeoMol | 44.6 | 41.4 | 0.875 | 0.834 | 43.0 | 36.4 | 0.928 | 0.841 |
| GeoDiff | 42.1 | 37.8 | 0.835 | 0.809 | 24.9 | 14.5 | 1.136 | 1.090 |
| Torsional Diffusion | 72.7 | 80.0 | 0.582 | 0.565 | 55.2 | 56.9 | 0.778 | 0.729 |
| TD w/ low temperature | | | | | | | | |
| - minimizing recall | 73.3 | 77.7 | 0.570 | 0.551 | 66.4 | 73.8 | 0.671 | 0.613 |
| - minimizing precision | 68.0 | 69.6 | 0.617 | 0.604 | 72.4 | 81.3 | 0.607 | 0.548 |
| TD w/ non-invariant PG | | | | | | | | |
| - minimizing recall | 75.8 | 81.5 | **0.542** | **0.520** | 66.2 | 72.4 | 0.668 | 0.607 |
| - minimizing precision | 58.9 | 56.8 | 0.730 | 0.746 | **76.8** | **88.8** | **0.555** | **0.488** |
| TD w/ invariant PG | | | | | | | | |
| - minimizing recall | **77.0** | **82.6** | 0.543 | **0.520** | 68.9 | 78.1 | 0.656 | 0.594 |
| - minimizing precision | 72.5 | 75.1 | 0.575 | 0.578 | 72.3 | 83.9 | 0.617 | 0.523 |

where $c$ symbolizes the text condition, $w \in \mathbb{R}^+$ is the guidance scale, and $\nabla_{\mathbf{x}_i} \log p_{t'}(\mathbf{x}_i, c) / \log p_{t'}(\mathbf{x}_i)$ is the conditional/unconditional scores, respectively. As probability ODE with classifier-free guidance is the prevailing method employed in text-to-image models [Saharia et al., 2022], we substitute the reverse-time SDE in Eq. (4) with the marginally equivalent ODE. Assuming that $f(\mathbf{x}_i, t') = 0$, the new backward ODE with particle guidance is

$$d\mathbf{x}_i = \left[ \frac{1}{2} g^2(t') \left( \tilde{s}(\mathbf{x}_i, c, t') - \alpha_{t'} \nabla_{\mathbf{x}_i} \sum_{j=1}^n k_{t'}(\mathbf{x}_i, \mathbf{x}_j) \right) \right] dt$$

Following SVGD [Liu & Wang, 2016], we employ RBF kernel $k_t(\boldsymbol{\tau}_i, \boldsymbol{\tau}_j) = \exp(-\frac{||\boldsymbol{\tau}_i - \boldsymbol{\tau}_j||^2}{h_t})$ with $h_t = m_t^2 / \log n$, where $m_t$ is the median of particle distances. We implement the kernel both in the original down-sampled pixel space (the latent of VAE) or the feature space of DINO-VIT-b/8 [Caron et al., 2021]. Defining the DINO feature extractor as $g_{\text{DINO}}$, the formulation in the feature space becomes:

$$d\mathbf{x}_i = \left[ \frac{1}{2} g^2(t') \left( \tilde{s}(\mathbf{x}_i, c, t') - \alpha_{t'} \nabla_{\mathbf{x}_i^0} \sum_{j=1}^n k_{t'} \left( g_{\text{DINO}}(\mathbf{x}_i^0), g_{\text{DINO}}(\mathbf{x}_j^0) \right) \right) \right] dt$$

where we set the input to the DINO feature extractor $g_{\text{DINO}}$ to the $\mathbf{x}_0$-prediction: $\mathbf{x}_i^0 = \mathbf{x}_i + \sigma(t')^2 \tilde{s}(\mathbf{x}_i, c, t')$, as $\mathbf{x}_0$-prediction lies in the data manifold rather than noisy images. $\sigma(t)$ is the standard deviation of Gaussian perturbatio kernel given time $t$ in diffusion models. The gradient w.r.t. $\mathbf{x}_i^0$ can be calculated by forward-mode auto-diff. We hypothesize that defining Euclidean distance in the feature space is markedly more natural and effective compared to the pixel space, allowing the repulsion in a more semantically meaningful way. Our experimental results in Fig. 6.1 corroborate the hypothesis.

To construct the data for evaluation, we randomly sample 500 prompts from the COCO validation set [Lin et al., 2014]. For each prompt, we generate a batch of four images. To get the average CLIP score/Aesthetic score versus in-batch similarity score curve, for I.I.D. sampling, we use $w \in \{6, 7.5, 8.5, 9\}$. We empirically observed that particle guidance achieved a much lower in-batch similarity score (better diversity) than IID sampling. As diversity typically improves with smaller guidance weights [Ho, 2022; Saharia et al., 2022], we chose a set of smaller guidance weights for I.I.D. sampling to further improve its diversity, keeping it in the relatively similar

range as particle guidance. Hence for particle guidance, we use a set of larger guidance scales: $w \in \{7.5, 8, 9, 9.5, 10\}$. Indeed, the experimental results suggest that even though I.I.D. sampling used a smaller guidance weight to promote diversity, its in-batch similarity score was still worse than that of particle guidance. . We set the hyper-parameter $\alpha_{t'}$ to $8\sigma(t)$ in particle guidance (feature) and $30\sigma(t)^2$ in particle guidance (pixel). We use an Euler solver with 30 NFE in all the experiments.

### E.2 IN-BATCH SIMILARITY SCORE

We propose in-batch similarity score to capture the diversity of a small set of generated samples $\{\mathbf{x}_1, \ldots, \mathbf{x}_n\}$ given a prompt $c$:

$$\text{In-batch similarity score}(\mathbf{x}_1, \ldots, \mathbf{x}_n) = \frac{1}{n(n-1)} \sum_{i \neq j} \frac{g_{\text{DINO}}(\mathbf{x}_i)^T g_{\text{DINO}}(\mathbf{x}_j)}{||g_{\text{DINO}}(\mathbf{x}_i)||_2 ||g_{\text{DINO}}(\mathbf{x}_j)||_2}$$

To save memory, we use the DINO-VIT-s/8 [Caron et al., 2021] as the feature extractor $g_{\text{DINO}}$.

## F EXTENDED IMAGE SAMPLES

In Fig. 8-Fig. 12, we visualize samples generated by the I.I.D. sampling process, particle guidance in the pixel space, and particle guidance in the DINO feature space, on four different prompts. For Fig. 8-Fig. 10, we select the prompts in Somepalli et al. [2023], with which Stable Diffusion is shown to replicate content directly from the LAION dataset. We also include the generated samples of SVGD-guidance, in which we replace the particle guidance term with SVGD formula (Eq. (2)). In Fig. 13, we observe that SVGD generally leads to blurry images when increasing the guidance scale $\alpha_t$. This is predictable as the guidance term in SVGD involves a weighted sum of scores of nearby samples, which will steer the samples toward the mean of samples.

Further, we provide at this URL `https://anonymous.4open.science/r/pg-images/` the (non-cherry picked) images generated from the first 50 text prompts of COCO using the same hyperparameters, the same random seed. We set the guidance scale to commonly used $w = 8$. When comparing visually the differences are subjective, but, to our eyes, many of the generations with PG show clear improvements in sample diversity. For example, for the first prompt "A baby eating a cake with a tie around his neck with balloons in the background.", conventional iid sampling tends to generate a brown-haired white child, while particle guidance generates diverse depictions of babies with varying hair and skin colors.

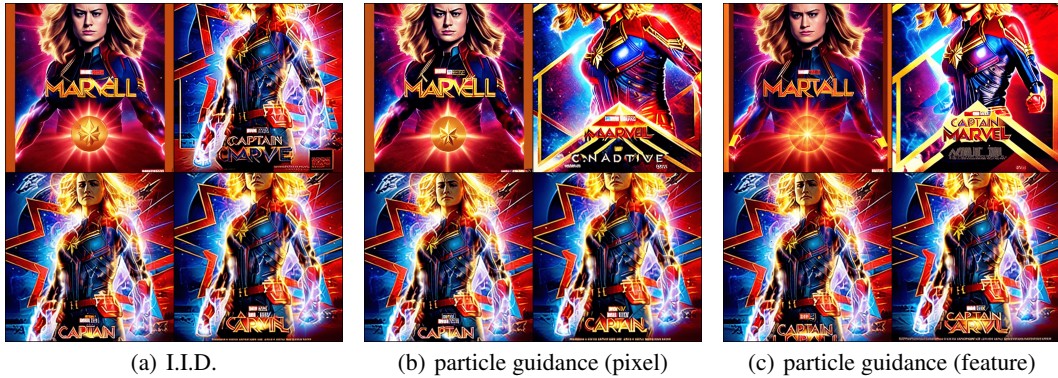

|  (a) I.I.D. | (b) particle guidance (pixel) | (c) particle guidance (feature) |

Figure 8: Text prompt: Captain Marvel Exclusive Ccxp Poster Released Online By Marvel

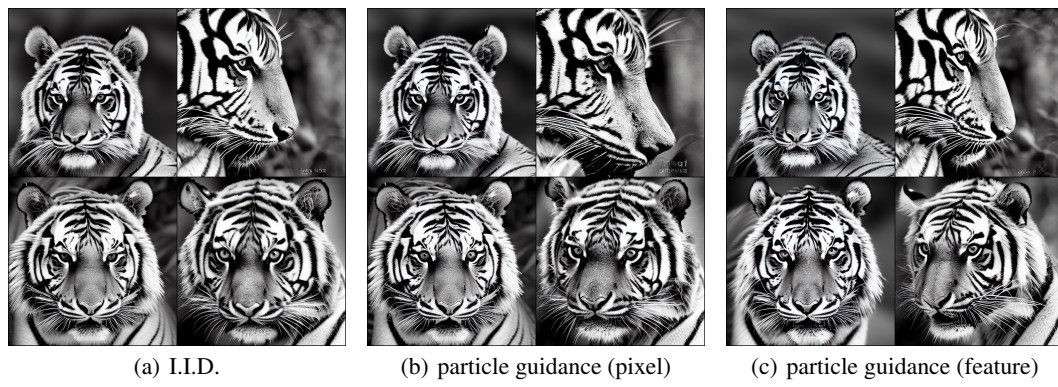

|  (a) I.I.D. | (b) particle guidance (pixel) | (c) particle guidance (feature) |

Figure 9: Text prompt: Portrait of Tiger in black and white by Lukas Holas

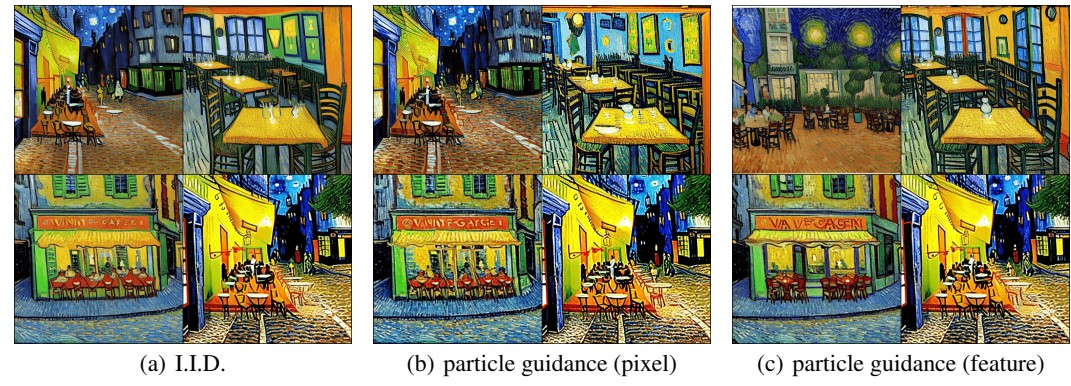

|  (a) I.I.D. | (b) particle guidance (pixel) | (c) particle guidance (feature) |

Figure 10: Text prompt: VAN GOGH CAFE TERASSE copy.jpg

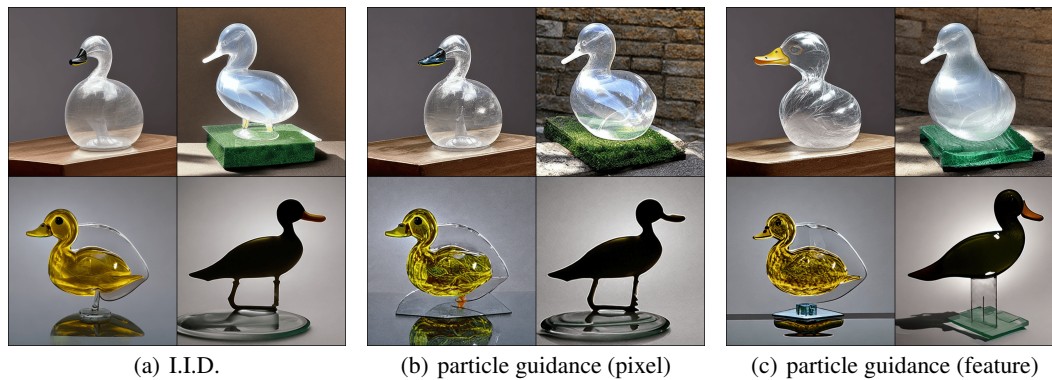

(a) I.I.D.        (b) particle guidance (pixel)        (c) particle guidance (feature)

Figure 11: Text prompt: A transparent sculpture of a duck made out of glass

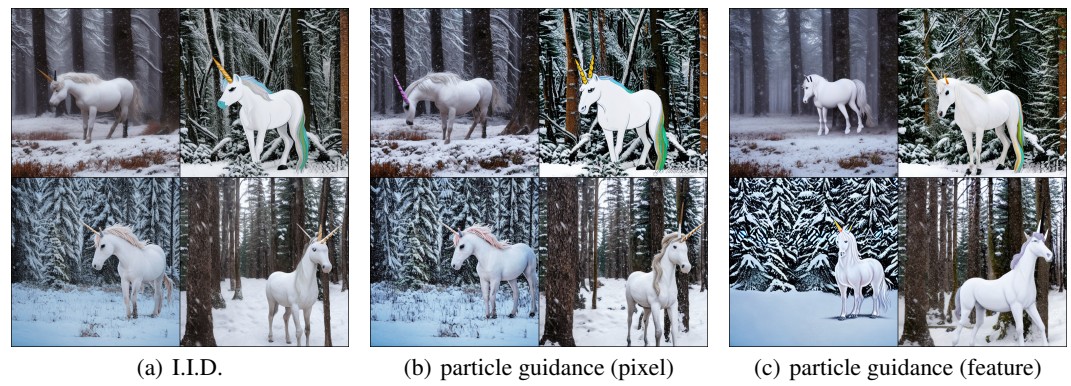

(a) I.I.D.        (b) particle guidance (pixel)        (c) particle guidance (feature)

Figure 12: Text prompt: A unicorn in a snowy forest

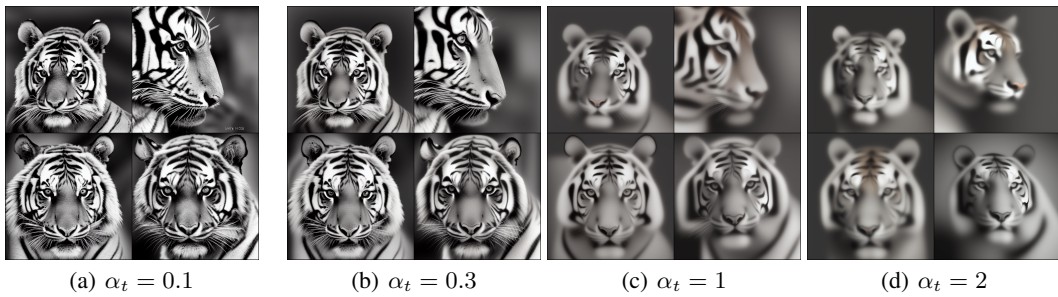

(a) $\alpha_t = 0.1$      (b) $\alpha_t = 0.3$      (c) $\alpha_t = 1$      (d) $\alpha_t = 2$

Figure 13: SVGD guidance, with varying $\alpha_t$

