# OpenReview forum: "Particle Guidance: non-I.I.D. Diverse Sampling with Diffusion Models"
_ICLR.cc/2024/Conference — ICLR 2024 poster_

### Official Review · Reviewer_AGeB · 2023-10-25

**Soundness:** 3 good
**Presentation:** 3 good
**Contribution:** 2 fair
**Rating:** 5
**Confidence:** 5

**Summary:**

This paper proposes the use of particle guidance for sampling in diffusion-based models, emphasizing that particle guidance increases diversity without reducing quality.

**Strengths:**

Diversity is a crucial property in generative modeling and sampling. Whether for multi-modal sampling or the regeneration of real data, we aim to cover each mode. The problem studied in this paper is significant.

The paper's approach is succinct and clear, making it easy to follow.

**Weaknesses:**

Despite the author's belief that diffusion models may suffer from mode collapse, previous experience suggests that mode collapse in diffusion models is not particularly severe [1]. If mode collapse is not a leading factor in most cases, it might affect the significance of this study. When the number of modes far exceeds the number of samples, it's evident that iid sampling and sampling with repulsion are similar, which seems to be the case for most data. For instance, in Figure 4 (a), diversity does not appear to be a severe issue. The main problem in Figure 4 (d) seems to be overfitting to the training data (an artificial setting). I think the authors should emphasize scenarios where the number of samples is close to the number of modes to highlight the setting's effectiveness. Molecular conformer might be an example, but since I am not an expert in that area, the authors could further explain.

The paper's theoretical foundation is weak and does not directly explain why particle guidance can increase diversity.


[1] A Study on Sample Diversity in Generative Models: GANs vs. Diffusion Models

**Questions:**

The authors are encouraged to further elucidate in what practical situations iid might fail (e.g., when the number of modes is close to the number of samples).

Has the paper investigated the impact of the number of particles on the final generation outcome?

Is there theoretical proof that particle guidance is better to some extent (even though this aligns with intuition)?

The paper uses an ODE solver. Would there be any changes if an SDE solver is used?

---

> ### Author Response · Authors · 2023-11-18
> **Response to Reviewer AGeB - Part 1/2**
>
> Thank you very much for your feedback and comments, we truly appreciate your time and effort. We have responded to each of your points below and integrated the responses in the manuscript.
>
> **Q: I think the authors should emphasize scenarios where the number of samples is close to the number of modes to highlight the setting's effectiveness. Molecular conformer might be an example, but since I am not an expert in that area, the authors could further explain. The authors are encouraged to further elucidate in what practical situations iid might fail (e.g., when the number of modes is close to the number of samples).**
>
> Thank you for raising this great point, we have added a subsection (B.1) to our appendix to discuss this point and refer to it in our main text. As suggested by the reviewer, when considering distributions with a number of modes, combinatorial analysis shows that IID sampling is particularly inefficient when the number of samples is in the same order as the number of modes. Intuitively, if the number of modes is much larger than the number of samples, then I.I.D. samples are still likely to capture separate modes; while if the number of samples is much larger than that of modes, the number of modes not captured by any sample is likely to be small.
>
> This is indeed the case in the setting of molecular conformer generation, where for each molecule the energy function has a finite set of relevant minima (called conformers) that one wants to find. In the commonly used DRUGS dataset, each molecule has on average ~100 conformers. Molecular conformer generation methods aim to recover all these modes with as few samples as possible (commonly for benchmarks the number of samples extracted from each method is twice the number of conformers). Therefore, molecular conformer generation falls into this setting where I.I.D. sampling is inefficient.
>
> We also discuss and point to other works showing the inefficiency of I.I.D. sampling when dealing with more general continuous distribution e.g. w.r.t. the variance of Monte Carlo estimation or improving diversity through determinantal point processes.
>
> Our experiments on text-to-image generation with stable diffusion also support the thesis that while typically there is no clear mode-collapse, diversity can still be improved. We provide at this URL https://anonymous.4open.science/r/pg-images/ the (non-cherry picked) images generated from the first 50 text prompts of COCO using the same hyperparameters and the same random seed (as recommended by another reviewer), with a fixed guidance scale $w=8$. Obviously when comparing visually the differences are subjective, but, at our eyes, many of the generations with PG show clear improvements in sample diversity. For example, for the first prompt “A baby eating a cake with a tie around his neck with balloons in the background.”, conventional I.I.D. sampling tends to generate a brown-haired white child, while particle guidance generates diverse depictions of babies with varying hair and skin colors.
>
>
> **Q: The paper's theoretical foundation is weak and does not directly explain why particle guidance can increase diversity. Is there theoretical proof that particle guidance is better to some extent (even though this aligns with intuition)?**
>
> Assuming that $\Phi$ is some measure of diversity then one can see particle guidance as locally improving the diversity of the samples at every step. However, for the general (fixed potential) setting it is hard to argue about global improvement or optimality as the effect on the final probability distribution depends on the data distribution itself (see Theorem 1). On the other hand, as we added to Section 7, using the maximum entropy theorem we can show that the learned potential particle guidance setup is globally optimal in the sense that it learns to sample the distribution minimizing the KL divergence with the independent distribution while satisfying a bound on the expected value of a diversity metric.

---

> ### Author Response · Authors · 2023-11-18
> **Response to Reviewer AGeB - Part 2/2**
>
> **Q: Has the paper investigated the impact of the number of particles on the final generation outcome?**
>
> Yes in our experiments we found that in general when one is interested in generating a certain number of samples in the end it is best (in terms of diversity) to use that as batch size or divide it in batches as large as possible. In the stable diffusion experiments, we use a batch size of 4, as this is the number of samples commonly reported. For the torsional diffusion experiments, where some molecules have a much higher number of conformers to be generated, we use larger batch sizes up to 128. We see monotonic improvements in performance when scaling the maximum batch size from 32 to 64 to 128 to 256. However, having larger batch sizes has two sources of runtime overhead, (1) the computation of the potential that, if parameterized as a sum of pairwise kernels scales quadratically with the batch size (although this is typically very fast compared to score model computation), and (2) the requirement of sampling all the elements of the batch in parallel that, for large batch sizes, requires moving elements in and out of GPU. These were the reasons we chose to clip the batch size to 128 for the torsional diffusion experiments: if the number of conformers to generate n is smaller or equal to 128 we use that, otherwise, we cover n with batches of size 128. We have added these discussions in the appendix of the manuscript.
>
> **Q: The paper uses an ODE solver. Would there be any changes if an SDE solver is used?**
>
> No, in fact for the synthetic and torsional diffusion experiments, we use an SDE solver. We clarified this point in the manuscript.
>
> ___
>
> Thank you again for the feedback, your comments have helped us to improve the manuscript. We hope that these improvements (see the general response for further additions) might persuade you of the contribution of our work and might warrant raising your score. We are happy to continue the conversation.

---

> > ### Comment · Reviewer_AGeB · 2023-11-22
> >
> > Thanks the authors feedback. However, I am still not fully convinced by the failure of iid setting for common generation. Indeed you can gain some diversity from the repulsive force, but I would regard it as a trick (which is very helpful for some artifical settings) at this moment.  Thus, I would maintain my rating.

---

> ### Author Response · Authors · 2023-11-22
>
> Thank you very much for the response and for continuing the discussion. We would argue, however, that it would be reductive to see our approach as the mere use of a repulsive force.
>
> What can be seen as a repulsive force is the gradient of a joint potential. The approach of using the gradient of this (time-evolving) potential to sample a joint distribution can be seen as a joint-particle equivalence of the score-based generative models for individual particles. In addition, unlike a vanilla approach to score-based modeling on sets of particles, the choice of the particular form of joint potential (e.g. see Eq. 5) and diffusion process has the advantages of (1) enabling the use of simple potentials with pretrained diffusion models (fixed potential particle guidance) and (2) not requiring access to samples of the joint distribution for training specific time-evolving potentials (learned potential particle guidance). As we now discuss in Section 7, following this joint gradient induces a distribution that optimizes the expected value of a (diversity) metric when controlling for some level of similarity (in terms of KL divergence) with the initial joint distribution.
>
> Further, in the experimental results, we consider two real tasks text-to-image and molecular conformer generation with non-artificial metrics and show that particle guidance can achieve better quantitative results than IID sampling. Especially, in the molecular conformer generation, a task of critical importance in many scientific domains, we were able, without retraining, to significantly improve the performance of the state-of-the-art on the most widely used benchmark and metrics. We believe this highlights both the failure of IID sampling and the efficacy of our approach.
>
> Finally, although this is the particular instantiation we focus on in the experimental results, the methodological and theoretical contributions of the paper can be applied beyond the "general diversity" objective. For example, one could define specific kernels to ensure a fair representation of different categories in generated samples (e.g. see above the discussion of images of babies by Stable Diffusion). As you mention in your review "diversity is a crucial property in generative modeling" and these are well-studied issues with existing models, we hope that particle guidance could become a new alternative framework to tackle these challenges.
>
> We have updated the manuscript to reflect this discussion and clarify these points. We hope this helps to convince you of the value of the contribution of our work and we are happy to continue the conversation.

---

### Official Review · Reviewer_ufFd · 2023-10-30

**Soundness:** 3 good
**Presentation:** 3 good
**Contribution:** 2 fair
**Rating:** 5
**Confidence:** 5

**Summary:**

The work investigates how to increase the diversity of one batch of samples for diffusion models. Specifically, the authors introduce a method named particle guidance, which is based on the gradient of a crafted time-varying potential field. The authors conduct various experiments to demonstrate the effectiveness of the method and provide an interesting analysis of the proposed method. However, some statements need to be clarified and more details about the experiments are needed.

**Strengths:**

- The topic, increasing diversity of generated samples, is important and crucial.
- The theoretical analysis conducted in the main paper and appendix are non-trivial and interesting.
- Experiments presented in the paper look good and show visual improvements.

**Weaknesses:**

Will the marginal distribution of Eq-1 be the same as that of the original diffusion model? It appears that the proposed method alters the marginal distribution. I am concerned that this shift in distribution may not be desirable in many applications.

It seems that most experiments conducted in this work focus on small batch sizes. I am interested in the authors' discussion regarding the scalability and effectiveness of the proposed method for large batch size.

There doesn't seem to be a principled approach to designing the proposed potential field besides the method present in Sec 6, which demands non-trivial training.

The illustrated plots in Figure 1 are a bit misleading to me. Why are the initial points concentrated in one mode? In a high-dimensional setting, the chance of sampling close-by Gaussian points is low.

Why were different guidance weights chosen for the IID and particle guidance experiments?

For fair evaluation purposes, could the authors post un-cherry-picked images on an anonymous website, for example, the first 50 text prompts of MS COCO or PartiPrompts? This should be done with the same hyper-parameters and random seed.

**Questions:**

See above

---

> ### Author Response · Authors · 2023-11-18
> **Response to Reviewer ufFd - Part 1/2**
>
> Thank you very much for your time and effort spent in reviewing and for the constructive feedback. We respond to all your comments and questions below and have integrated these additions to the manuscript.
>
> **Q: Will the marginal distribution of Eq-1 be the same as that of the original diffusion model? It appears that the proposed method alters the marginal distribution. I am concerned that this shift in distribution may not be desirable in many applications.**
>
> Thank you for raising this point. Indeed the marginal distribution of the sampled points under the particle guidance is different from the original data distribution. We discuss this question of preservation of marginal distribution in the manuscript (see for example Section 7.2).
>
> We note, that this (lack of preservation of the marginals) is the case also in all the other non-IID sampling methods discussed in related works (Section 4) and is, arguably, positive depending on the definition of optimality (see new components in Section 7). However, we do recognize that there are several domains where preserving marginal distribution is desirable (e.g. if using samples for some estimates). Therefore, in the setting of learned potential PG, we derive a training procedure (Section 7.2) for the joint potential that preserves marginal distributions (still in a maximum entropy optimal way).
>
>
> **Q: It seems that most experiments conducted in this work focus on small batch sizes. I am interested in the authors' discussion regarding the scalability and effectiveness of the proposed method for large batch size.**
>
> Thank you for the suggestion we added a discussion on the scalability of the method to large batch sizes in Appendix B.2 and better clarification of the batch size that we use in each experiment. While in the stable diffusion experiments, we use a batch size of 4, for the torsional diffusion experiments we use significantly larger batch sizes up to 128. We see monotonic improvements in performance when scaling the maximum batch size from 32 to 64 to 128 to 256. However, as now explained in the new Appendix having larger batch sizes has two sources of runtime overhead, (1) the computation of the potential that, if parameterized as a sum of pairwise kernels scales quadratically with the batch size (although this is typically very fast compared to score model computation), and (2) the requirement of sampling all the elements of the batch in parallel that, for large batch sizes, requires moving elements in and out of GPU. These were the reasons we chose to clip the batch size to 128 for the torsional diffusion experiments.
>
> **Q: There doesn't seem to be a principled approach to designing the proposed potential field besides the method present in Sec 6, which demands non-trivial training.**
>
> Indeed, as mentioned in Sec 7, in order to precisely control the joint distribution one cannot rely on simple time-evolving potentials but these have to depend on the data distribution and therefore learned from data. However, we do not see this necessarily as a weakness for multiple reasons. (1) Theorem 1 shows that even with simple kernels one can affect the joint distribution in a way that encourages diversity. (2) We argue that in most domains precisely controlling the joint distribution is not a requirement, for example, in the first prompt of the images requested below the exact distribution of babies is not important while improving diversity often is. Moreover, the possibility to define tailored kernels only on certain projections of the data could enable direct control of the diversity on features of interest (e.g. ethnicity, gender, …). (3) The fact that there are no trivial solutions to this problem is a further reason why we think that studying it is important for the field. In Section 6 (now Section 7), we go in even further detail in our analysis deriving both training schemes and optimality guarantees of the learned distribution. In general, we hope that our work can spark increased interest in this problem in the field.
>
> **Q: The illustrated plots in Figure 1 are a bit misleading to me. Why are the initial points concentrated in one mode? In a high-dimensional setting, the chance of sampling close-by Gaussian points is low.**
>
> Figure 1 tries to illustrate the potential failure cases of IID sampling to give an intuition of why particle guidance can help. Indeed for a distribution with three equiprobable modes, the likelihood that only one mode is discovered by 3 IID samples is 1/9 (but that all three are discovered is also only 2/9), however, in this case, it does not depend on the dimensionality of the space but only on the number of modes. We have added a more careful analysis of the failure modes of IID sampling in Appendix B.1 and reference it in the Figure 1 caption.

---

> ### Author Response · Authors · 2023-11-18
> **Response to Reviewer ufFd - Part 2/2**
>
> **Q: Why were different guidance weights chosen for the IID and particle guidance experiments?**
>
> Thank you for pointing this out. We observed that particle guidance achieved a much lower in-batch similarity score (better diversity) than IID sampling. As diversity typically improves with smaller guidance weights [1, 2], we chose a set of smaller guidance weights for i.i.d. sampling to further improve its diversity, keeping it in the relatively similar range as particle guidance. Indeed, the experimental results suggest that even though IID sampling used a smaller guidance weight to promote diversity, its in-batch similarity score was still worse than that of particle guidance.
>
> [1] Photorealistic Text-to-Image Diffusion Models with Deep Language Understanding, Saharia et al, NeurIPS 2022
>
> [2] Classifier-free diffusion guidance, Ho et al, arXiv:2207.12598
>
> **Q: For fair evaluation purposes, could the authors post un-cherry-picked images on an anonymous website, for example, the first 50 text prompts of MS COCO or PartiPrompts? This should be done with the same hyper-parameters and random seed.**
>
> Thank you for the suggestion. As requested we provide at this URL https://anonymous.4open.science/r/pg-images/ the images generated from the first 50 text prompts of COCO using the same hyperparameters and the same random seed and we added this URL to the manuscript as well. We set the guidance scale to commonly used $w=8$. Obviously when comparing visually the differences are subjective, but, at our eyes, many of the generations with PG show clear improvements in sample diversity. For example, for the first prompt “A baby eating a cake with a tie around his neck with balloons in the background.”, conventional I.I.D. sampling tends to generate a brown-haired white child, while particle guidance generates diverse depictions of babies with varying hair and skin colors.
>
> ___
>
> Thank you again for your suggestions and constructive feedback! We hope that the responses above and the various improvements to the manuscript that stemmed from them (see the general response for further improvements) might convince you to raise your score. We are happy to continue the conversation.

---

### Official Review · Reviewer_rfgv · 2023-10-31

**Soundness:** 4 excellent
**Presentation:** 4 excellent
**Contribution:** 3 good
**Rating:** 8
**Confidence:** 4

**Summary:**

This paper presents a method to sample from a joint distribution of a diffusion model combined with some potential function, with a specific focus on potentials that encourage diversity. This is done by attaching an additional "guidance" term to the diffusion SDE that moves the generation towards high-potential region, which shares the similar idea of prior works on guidance, e.g. classifier guidance. The paper analyzes the theoretical properties of the implied joint distribution, and connections to related works. Empirically, the paper demonstrates the superiority of the proposed method in Gaussian synthetic example, text-to-image generation and molecular conformation generation. In these examples, the proposed method has scored better in diversity and other downstream metrics compared to the standard IID samping.

**Strengths:**

- The paper is very well-written and easy to follow.

- The motivation of this paper is very clear and seems to be important.

- The proposed method presents simple and effective way to overcome the limitation of IID sampling, which is the inefficiency in exploring all possible modes.

- The empirical comparison is comprehensive and convincing.

**Weaknesses:**

- While the angle of this paper (to promote efficiency and diversity) is novel, I found the methodology itself, namely equation 1, does not have too much _technical_ novelty, and seems to be following the line of "guidance" works, e.g. classifier guidance, reconstruction guidance [1], DPS[2], Pseudo-inverse diffusion sampler [3], universal guidance [4], etc.

- One potential weakness is the memory constraint invoked by simulating multiple particles (and getting their gradients) at the same time. Can the authors comment on this point?

- Another missing discussion is on the choice of $n$, the number of particles used in practice. How does one determine $N$, and what is its implication on the joint distribution? For example, I hypothesize that for $n_1>n_2$, marginalizing out the last $n1-n2$ particles in the joint distribution of $n1$ does not recover the joint distribution of $n_2$. And when $n$ is much larger than the number of modes in the diffusion model that we are interested in, what would happen by running the proposed method (especially when using a large guidance weight). Would that result in generating particles that have very low density under the diffusion model?

- This point is regarding clarify. Similar to the above "marginalization" argument, I think there is a missing discussion on whether such joint sampling of $x_1, \cdots, x_n$ would recover marginally exact samples from a diffusion model, and I assume not (correct me if I'm wrong). But somehow I found the paper can be misleading in suggesting "yes", e.g. on page 3 "Intuitively, this will push our different samples to be dissimilar from one another while at the same time matching our distribution, improving sample efficiency."

[1] Video Diffusion Models. Jonathan Ho, Tim Salimans, Alexey Gritsenko, William Chan, Mohammad Norouzi, David J. Fleet
https://arxiv.org/abs/2204.03458
[2] Diffusion Posterior Sampling for General Noisy Inverse Problems. Hyungjin Chung, Jeongsol Kim, Michael T. Mccann, Marc L. Klasky, Jong Chul Ye
https://arxiv.org/abs/2209.14687
[3] Pseudoinverse-Guided Diffusion Models for Inverse Problems. Jiaming Song, Arash Vahdat, Morteza Mardani, Jan Kautz
https://openreview.net/forum?id=9_gsMA8MRKQ
[4] Universal Guidance for Diffusion Models. Arpit Bansal, Hong-Min Chu, Avi Schwarzschild, Soumyadip Sengupta, Micah Goldblum, Jonas Geiping, Tom Goldstein
https://arxiv.org/abs/2302.07121

**Questions:**

1. In abstract, can you explain which part of the paper discusses "its implications on the choice of potential"?

2. The "finite-sampling" property of diffusion models does not seem to be accurate. Apparently, sampling from the diffusion SDE (in a discrete manner, and without running infinite Langevin steps) will also accumulate error. It seems like the authors were referring this property to getting to every "mode" of the distribution in finite steps. If this is the case, I wonder the authors can be more clear about this point, and provide evidence or reference to support this claim?

3. Can you further explain why "Hence the density p0ˆ can be understood as a reweighting of the random variable Z" (the sentence under equation 5 on page 4)? Furthermore, can you provide intuition on what the random variable $Z$ encode?

4. Maybe I missed this part, but for all experiments, what are the $n$, number of particles you used, and how do you determine them?

5. In the preamble part of section 5, the reference to the text-to-image experiment is missing.

6. In Figure 3, how is "varying guidance scale from 6 to 10" reflected/used in the figure and this experiment?

7. Table 1 seems to miss the reference and descriptions of competing methods. Furthermore, I wonder why the authors didn't experiment the proposed method on geodiff model. Is that possible?

8. Section 6 seems really interesting! I wonder if there is any practical challenge in instantiating that paradigm?

---

> ### Author Response · Authors · 2023-11-18
> **Response to Reviewer rfgv - Part 1/3**
>
> Thank you very much for your thorough review, we truly appreciate your time and effort. We responded below to all your comments and questions and adapted the manuscript to reflect them.
>
> **Q: "I found the methodology itself, namely equation 1, does not have too much technical novelty, and seems to be following the line of "guidance" works"**
>
> Although we chose to present our method from the perspective of guidance to make it as easy as possible to understand for researchers in the field, we believe that our methodology is clearly distinct from such previous works. In particular, we believe our work presents novel ideas w.r.t. this line of guidance works on several fundamental aspects: (1) the goal and motivation, (2) the potential being applied on a set of particles instead of individual ones (3) the form of the potential analyzed (either fixed or learned) (4) the theoretical analysis of effects and (5) the connections to methods in other disciplines.
>
> **Q: "One potential weakness is the memory constraint invoked by simulating multiple particles (and getting their gradients) at the same time. Can the authors comment on this point? Another missing discussion is on the choice of the number of particles used in practice."**
>
> Thank you for raising this point, we have now added this discussion in Appendix B.2 of the paper. In terms of memory, particle guidance does not create any significant overhead since the kernel computations can be aggregated per element. However, when running inference on GPU if n is larger than the batch size that fits the GPU memory when running score model inference, the data might have to be moved back and forth between RAM and GPU memory to enable synchronous steps for particle guidance, causing runtime overhead.
>
> In the case of our experiments on Stable Diffusion, the number of samples extracted (4) does not create significant overhead. However, in the setting of conformer generation on DRUGS different molecules can have very different numbers of conformers with some even having thousands of them. For efficiency, we therefore cap the size of n in particle guidance to 128 and perform batches of 128 samples until the total number of conformers is satisfied.
>
> **Q: "...Would that result in generating particles that have very low density under the diffusion model?" … I think there is a missing discussion on whether such joint sampling of would recover marginally exact samples from a diffusion model"**
>
> Thank you for raising this point. Indeed the marginal distribution of the sampled points under the particle guidance is different from the original data distribution. We fixed the comment on page 3 and now discuss this question of preservation of marginal distribution in the manuscript (see for example Section 7.2).
>
> We note, that this (lack of preservation of the marginals) is the case also in all the other non-IID sampling methods discussed in related works (Section 4) and is, arguably, positive depending on the definition of optimality (see new components in Section 7). However, we do recognize that there are several domains where preserving marginal distribution is desirable (e.g. if using samples for some estimates). Therefore, in the setting of learned potential PG, we derive a training procedure (Section 7.2) for the joint potential that preserves marginal distributions (still in a maximum entropy optimal way).
>
> **Q1: In abstract, can you explain which part of the paper discusses "its implications on the choice of potential"?**
>
> In both the synthetic and the text-to-image settings we test and analyze different choices of potentials (full-dimensional vs submanifold, pixel-space vs feature-space). However, to avoid possible confusion we have removed this sentence from the abstract.

---

> ### Author Response · Authors · 2023-11-18
> **Response to Reviewer rfgv - Part 2/3**
>
> **Q2: The "finite-sampling" property of diffusion models does not seem to be accurate. Apparently, sampling from the diffusion SDE (in a discrete manner, and without running infinite Langevin steps) will also accumulate error…I wonder the authors can be more clear about this point, and provide evidence or reference to support this claim?**
>
> Thank you for raising this important point. In general, for the same order of discretization error, diffusion SDE can efficiently sample from data distribution in much fewer steps than Langevin dynamics. For instance, Theorem 1 of [1] shows that, assuming accurate learning of score, convergence of diffusion SDE is independent of isoperimetry constant of target distribution. Langevin dynamics mixing speed can be exponentially slow if spectral gap/isoperimetry constant is small.
>
> The sampling process in diffusion models essentially involves solving an ODE/SDE, which contrasts with the Langevin dynamics in energy-based models or the SVGD. With significantly fewer sampling steps, diffusion models can achieve a much smaller distance between the generated distribution and the real distribution. For instance, Theorem 1 in [2] shows that the Wasserstein-1 distance between the generated and real distributions in a diffusion SDE can be upper bounded by $O(\sqrt{\delta}T+\epsilon_{approx})T^2$, where $\delta$ is the step size, $\epsilon_{approx}$ is the estimation errors of score function of neural network, and $T$ is the interval length of diffusion SDE (from time $T$ to time 0). This suggests that diffusion models can effectively cover the true distribution with finite steps, efficiently reaching various distribution modes. This is also empirically confirmed by recent diffusion models which can generate good samples in just tens of steps [3].
>
> In contrast, Langevin dynamics exhibits slow mixing times, which increase exponentially with data dimensionality, to attain adequate coverage of the true distribution [4]. Empirically, one also observes that Langevin dynamics need to take thousands of iterations to generate clean images [5]. In a similar vein, SVGD struggles to identify all isolated modes within a finite time [6].
>
> We’ve clarified this point in our revised version.
>
> [1] Sampling is as easy as learning the score: theory for diffusion models with minimal data assumptions, Chen et al, arxiv: 2209.11215
>
> [2] Restart Sampling for Improving Generative Processes, Xu et al, NeurIPS 2023
>
> [3] Elucidating the Design Space of Diffusion-Based Generative Models, Karras et al, NeurIPS 2022
>
> [4] Non-Convex Learning via Stochastic Gradient Langevin Dynamics: A Nonasymptotic Analysis, Raginsky et al, Proceedings of Machine Learning Research vol 65:1–30, 2017
>
> [5] Generative Modeling by Estimating Gradients of the Data Distribution, Song et al, NeurIPS 2019
>
> [6] Blindness of score-based methods to isolated components and mixing proportions, Li et al, arxiv: 2008.10087
>
>
> **Q3: Can you further explain why "Hence the density p0ˆ can be understood as a reweighting of the random variable Z" (the sentence under equation 5 on page 4)? Furthermore, can you provide intuition on what the random variable Z encode?**
>
> Indeed the derived expression in Theorem 1 is not very interpretable, in Appendix A.1 we tried to add some intuition for the final expression, however, we agree that the joint potential for arbitrary fixed potential particle guidance is hard to interpret. On the other hand, we added further intuition to the joint potential learned when applying the learned potential particle guidance (see the general response) as the distribution minimizing the KL divergence with the independent distribution while satisfying a bound on the expected value of a (diversity) metric.
>
> **Q4: Maybe I missed this part, but for all experiments, what is number of particles you used, and how do you determine them?**
>
> For the Stable Diffusion experiments n=4, as this is what is returned in practice in most text-to-image generation tools. For the molecular conformer generation experiments, where the number of conformers one has to generate for every molecule is given by the benchmark (often ~100-200 but it can vary significantly), we used batches of up to n=128 until all the conformers for that particular molecule are generated. We clarified this point in the text.
>
> **Q5: In the preamble part of section 5, the reference to the text-to-image experiment is missing.**
>
> Thank you for raising this point. We fixed the relevant paragraph.

---

> ### Author Response · Authors · 2023-11-18
> **Response to Reviewer rfgv - Part 3/3**
>
> **Q6: In Figure 3, how is "varying guidance scale from 6 to 10" reflected/used in the figure and this experiment?**
>
> Thank you for pointing this out. We are using classifier-free guidance [1], which allows us to control the diversity, CLIP score, and Aesthetic score of the generated images by varying the guidance scale. As shown in many previous works [2, 3, 4, 5], a higher guidance scale typically results in better text-image alignment and fidelity (aesthetic score), but lower diversity (FID score and in-batch similarity score).
> In this experiment, we qualitatively study how different methods balance diversity against text-image alignment/fidelity by varying the guidance scale from 6 to 10. As shown in Figure 3, the CLIP score (higher is better) and Aesthetic score (higher is better) improve, while the in-batch similarity score (lower is better) degrades, as we increase the guidance scale in classifier-free guidance. Figure 3 also shows that our proposed particle guidance better balances diversity and text-image alignment/fidelity than i.i.d. sampling, achieving strictly better diversity for the same CLIP/aesthetic score.
> We’ve added more discussions about the role of guidance scale in our updated version.
>
> [1] Classifier-free diffusion guidance, Ho et al, arXiv:2207.12598
>
> [2] On Distillation of Guided Diffusion Models, Meng et al, CVPR 2023
>
> [3] Photorealistic Text-to-Image Diffusion Models with Deep Language Understanding, Saharia et al, NeurIPS 2022
>
> [4] Restart Sampling for Improving Generative Processes, Xu et al, NeurIPS 2023
>
> **Q7: Table 1 seems to miss the reference and descriptions of competing methods. Furthermore, I wonder why the authors didn't experiment the proposed method on geodiff model. Is that possible?**
>
> Thank you for the suggestion, we have added a short description of each baseline alongside their references in Appendix D and linked to it in Table 1 caption. While particle guidance could definitely be applied also to GeoDiff, the reasons why we chose to test it on torsional diffusion instead are: (1) torsional diffusion has shown better performance on the task (2) GeoDiff due to its many steps has a reported inference time almost two orders of magnitude slower [Jing et al. 2022] (3) because torsional diffusion operates on a much lower dimensional space we believe that particle guidance can have a larger impact.
>
> **Q8: Section 6 seems really interesting! I wonder if there is any practical challenge in instantiating that paradigm?**
>
> Compared to the fixed potential particle guidance the main practical challenge is the need to define a neural architecture operating on a set of samples and train a model for the potential. We believe this could find use in scientific applications where controlling for the resulting joint or marginal distribution is important. However, in many current applications of diffusion models, like the ones presented in the experiments section, this does not seem to be a requirement.
>
> ___
>
> Thank you again for the thoughtful feedback and questions! We hope that the clarifications above combined with the various improvements of the manuscript that stemmed from them (see the general response for further improvements) will strengthen your positive opinion on the significance of the contribution of our work!

---

> > ### Comment · Reviewer_rfgv · 2023-11-19
> >
> > I would like to thank the authors for their detailed response. Most of my concerns are addressed and I will raise my score.

---

> > > ### Author Response · Authors · 2023-11-21
> > >
> > > Thank you very much for the response!

---

### Official Review · Reviewer_iSmF · 2023-10-31

**Soundness:** 3 good
**Presentation:** 4 excellent
**Contribution:** 3 good
**Rating:** 6
**Confidence:** 4

**Summary:**

The aim of the methodology developed in this paper is to promote sample diversity when sampling I.I.D. from a diffusion model. Indeed, when drawing a $M$ i.i.d. samples from a multimodal density with $M$ modes, it is unlikely to get a sample on each one of the modes, even when all the modes have the same weight. To promote sample diversity when sampling from a diffusion model, the authors propose to modify the backward process by adding a repulsion term ensuring that samples drawn at each step of the diffusion process are as dissimilar as possible.

**Strengths:**

1- The problem that this paper tries to solve is quite original.
2- The paper is well written and the idea in itself is interesting. It is also quite nice that authors managed to give an explicit formula for the the joint density targeted by their modified backward process. The connection with other works that make use of a repulsion term is also a nice addition.
3- The experiments are well explained and sound.

**Weaknesses:**

1- While Theorem 1 is interesting, the expression derived for the joint density is not very interpretable and so one does not get a good grasp of what the modified backward process is targeting. It is quite unsatisying that the authors did not add a toy experiment where they explicitely compare the law of the samples obtained from particle guidance with the initial law that is targeted. I believe that the authors should consider an experiment in which the modes do not have the same weights and then show what is actually the law that they are sampling from. I would expect this law to not have the correct statistical weights, which could be quite unconvenient.

2- Failure cases of the proposed method are not discussed. Furthermore, as far as I can tell there is no real discussion on the choice of the potential and its parameters (besides in the toy example). How did the authors choose the potential in the other examples? Can a badly chosen potential perform worse than the original diffusion model?

**Questions:**

Why did the authors choose such a small variance for the Gaussian mixture experiment?

---

> ### Author Response · Authors · 2023-11-18
> **Response to Reviewer iSmF**
>
> Thank you very much for the thoughtful review, we truly appreciate your time and effort. We responded below to all your comments and questions and adapted the manuscript to reflect them.
>
> **Q: the expression derived for the joint density is not very interpretable and so one does not get a good grasp of what the modified backward process is targeting.**
>
> Indeed the derived expression in Theorem 1 is not very interpretable, in Appendix A.1 we tried to add some intuition for the final expression, however, we recognize that the joint potential for arbitrary fixed potential particle guidance is hard to interpret. On the other hand, we added further intuition to the joint potential learned when applying the learned potential particle guidance (see the general response) as the distribution minimizing the KL divergence with the independent distribution while satisfying a bound on the expected value of a (diversity) metric.
>
> **Q: I would expect this law to not have the correct statistical weights, which could be quite unconvenient.**
>
> Thank you for raising this point. Indeed the marginal distribution of the sampled points under the particle guidance is different from the original data distribution. As now discussed in the paper (section 7.2 and appendix A.3), this is the case also in all the other non-IID sampling methods discussed in related works (Section 4) and is, arguably, positive depending on the definition of optimality (see new components in Section 7). However, we do recognize that there are several domains where preserving marginal distribution is desirable (e.g. if using samples for some estimates). Therefore, in the setting of learned potential PG, we derive a training procedure (Section 7.2) for the joint potential that preserves marginal distributions (still in a maximum entropy optimal way).
>
> **Q: Failure cases of the proposed method are not discussed. Furthermore, as far as I can tell there is no real discussion on the choice of the potential and its parameters (besides in the toy example). How did the authors choose the potential in the other examples? Can a badly chosen potential perform worse than the original diffusion model?**
>
> Thank you for the suggestion. We tried to clarify these points in the manuscript and added discussions in Appendix B on the failure cases of particle guidance and when this can provide benefits over I.I.D. sampling. We discuss and benchmark the different choices of potential both in Sections 5.1 (synthetic experiments) and 5.2 (text-to-image experiments). Indeed a badly chosen potential or in general one with a guidance weight too high can overly change the marginal likelihood and negatively impact the sampling quality (see Figure 5). However, in fixed potential particle guidance, its parameters can be easily fit at inference time, therefore, it is typically relatively inexpensive to test the optimal value of the guidance weight for the application of interest and the chosen potential. This leads to the prevention of too high guidance weights and the simple detection of bad potential when the optimal weight is close to 0.
>
> **Q: Why did the authors choose such a small variance for the Gaussian mixture experiment?**
>
> We chose a small variance in the Gaussian mixture experiment to have separate modes in the distribution that could make the mode recovery analysis possible. While one can also analyze diversity and coverage in different ways for "more homogeneous" distributions, we believe this perspective makes it easier to understand. Moreover, it is closely related to the setup of conformer generation where molecules are characterized by a finite number of somewhat distinct possible conformations (conformers).
>
> ___
>
> Thank you again for the feedback! We hope that the clarifications above combined with the various improvements of the manuscript that stemmed from them (see the general response for further improvements) will strengthen your positive opinion on the significance of the contribution of our work!

---

### Author Response · Authors · 2023-11-18
**General response**

We would like to thank all the reviewers for their thoughtful reviews, we appreciate the time taken to analyze the paper carefully, the feedback was very useful and pushed us to make a number of improvements to the manuscript. We summarize the main changes below and then respond to each individual point in separate responses.

Firstly, to make space for the new contributions discussed below and provide a more clear presentation of the method we slightly rearranged some of the sections of the manuscript and now make a direct distinction between fixed potential particle guidance and learned potential particle guidance. The former provides ready-to-use potentials that require no further training and have little inference overhead, and the latter requires a training process but offers theoretical guarantees.

Some of the reviewers raised concerns about the marginal distribution of the generated data being modified under particle guidance. As now discussed in the paper (both section 7.2 and appendix A.3), this is the case also in all the other non-IID sampling methods discussed in related works (Section 4, e.g. [1,2]). Furthermore, this is, arguably, positive depending on the definition of optimality: the distribution that we learn is the one satisfying a bound on the expected value of a (diversity) metric $\Phi_0$ while minimizing the KL divergence with the independent distribution (see new components in Section 7). However, we do recognize that there are several domains where preserving marginal distribution is desirable (e.g. if using samples for some estimates). Therefore, in the setting of learned potential particle guidance, we derive a training procedure (Section 7.2) for the joint potential that preserves marginal distributions (still in a maximum entropy optimal way).

We also strengthened our theoretical analysis. In Appendix B.1, we discuss in more detail the suboptimality of I.I.D. sampling from different standpoints. Furthermore, we now provide a more founded motivation for the learned potential in Section 7: the learned distribution is the one minimizing the KL divergence with the independent distribution while satisfying a bound on the expected value of a (diversity) metric.

As suggested by one of the reviewers, we provide at this URL https://anonymous.4open.science/r/pg-images/ the (non-cherry picked) images generated from the first 50 text prompts of COCO using the same hyperparameters and the same random seed, with a fixed classifier-free guidance scale $w=8$. Obviously when comparing visually the differences are subjective, but, at our eyes, many of the generations with PG show clear improvements in sample diversity. For example, for the first prompt “A baby eating a cake with a tie around his neck with balloons in the background.”, conventional I.I.D. sampling tends to generate a brown-haired white child, while particle guidance generates diverse depictions of babies with varying hair and skin colors.

We hope that these additions can clarify the components of the manuscript you found unclear and strengthen your consideration of its contributions.

[1] Liu, Qiang, and Dilin Wang. "Stein variational gradient descent: A general purpose bayesian inference algorithm." Advances in neural information processing systems 29 (2016).

[2] Laio, Alessandro, and Michele Parrinello. "Escaping free-energy minima." Proceedings of the national academy of sciences 99.20 (2002): 12562-12566.

---

### Meta-Review · Area_Chair_J8iP · 2023-12-10

**Metareview:**

Most of the reviewers are really ambivalent about this paper. It examines an interesting idea of promoting diverse exploration among the samples that the diffusion model generates. On the other hand, it is not crystal clear how much benefit that brings into the literature of diffusion models. The theoretical analyses also falls short of explaining that. I'm leaning towards accepting the paper, given that in certain scenarios, perhaps not the ones considered in this paper, such idea may become useful.

**Justification For Why Not Higher Score:**

It is not very clear how much benefit enforcing diverse exploration brings into the literature of diffusion models, since in practice, the problem of mode collapse is not so prominant in diffusion models.

**Justification For Why Not Lower Score:**

I'm leaning towards accepting the paper, given that in certain scenarios, perhaps not the ones considered in this paper, such idea may become useful.

---

### Decision · Program_Chairs · 2024-01-16

Accept (poster)